# Overcoming laser phase noise for low-cost coherent optical communication

Xiansong Fang[1,4], Yixiao Zhu [2,4] ✉, Xiang Cai[1], Weisheng Hu[3], Zhixue He[3], Shaohua Yu [1,3] & Fan Zhang[1,3] ✉

Artificial-intelligence-generated content has driven explosive data traffic growth in data-center interconnects. Traditional direct detection solutions struggle with limited spectral efficiency and distance, prompting the shift to coherent optics for cost-sensitive short-reach links. One specific challenge is integrating low-cost lasers while overcoming severe phase noise on high-order modulation formats. Here, we propose a residual carrier modulation scheme for precise and efficient carrier frequency and phase recovery. The residual optical carrier can continuously track phase fluctuations without redundancy compared with discrete time-domain pilots, and address the digital-to-analog convertor resolution reduction issue of frequency-domain digital pilots. In proof-of-concept experiments, we transmit a net 1-Tb/s probabilistic-shaped 256-ary quadrature amplitude modulated (PS-256-QAM) signal using a 3 MHz distributed feedback (DFB) laser. Our scheme improves bitrate by 41% compared to conventional time-domain pilots, achieving a record laser linewidth sum and symbol duration product of $6.89 \times 10^{-5}$. This approach supports MHz linewidth DFB lasers in low-cost coherent optical communications.

The modern information society relies on the optical fiber infrastructure, which is responsible for >99% of global data transmission and exchange. As the era of emerging artificial intelligence-generated content (AIGC) unfolds, significant volumes of data are generated at the cloud side and distributed to end-users. Consequently, there is evident traffic growth in the short-reach scenarios (Fig. 1a), including data-center interconnects[1,2] and fiber-wireless access networks[3,4]. Different from long-haul transmission[5,6], short-reach scenarios have abundant fiber resources, allowing for massively parallel deployment[7,8]. Hence, there is a stringent requirement for cost-effective transceivers, particularly the laser source.

The optical transmission systems can be divided into two categories: direct detection and coherent detection[9]. The traditional direct detection solution employs only one-dimensional power detection, whereas coherent detection can fully utilize the phase and polarization diversity based on a reference laser, namely the local oscillator (LO).

However, as the signal laser and LO originate from different sources, the phase noise becomes a critical impairment for coherent receivers[10]. Theoretically, laser phase noise is modeled as a Wiener process[11], and its variance is determined by the product of the symbol duration and the linewidth sum of signal laser and LO.

As the Ethernet interface is speeding up from 800GbE to 1.6TbE and beyond, the adoption of multi-level modulation formats emerges as a promising solution to overcome the bandwidth limitation of optoelectronic devices and components[12]. For decades, intensity-modulation with direct detection (IM-DD) scheme in Fig. 1b has dominated optical interconnects, thanks to its cost and simplicity advantages. However, the square-law detection of the photodiode imposes fundamental restrictions. More specifically, the line rate is mainly limited by amplitude-only one-dimensional modulation, and the distance is hindered by the fiber chromatic dispersion-induced power fading effect, primarily due to the loss of phase information[13].

---

[1]State Key Laboratory of Advanced Optical Communication Systems and Networks, Frontiers Science Center for Nano-optoelectronics, School of Electronics, Peking University, Beijing 100871, China. [2]State Key Laboratory of Advanced Optical Communication Systems and Networks, Department of Electronic Engineering, Shanghai Jiao Tong University, Shanghai 200240, China. [3]Peng Cheng Laboratory, Shenzhen 518055, China. [4]These authors contributed equally: Xiansong Fang, Yixiao Zhu. ✉e-mail: yixiaozhu@sjtu.edu.cn; fzhang@pku.edu.cn

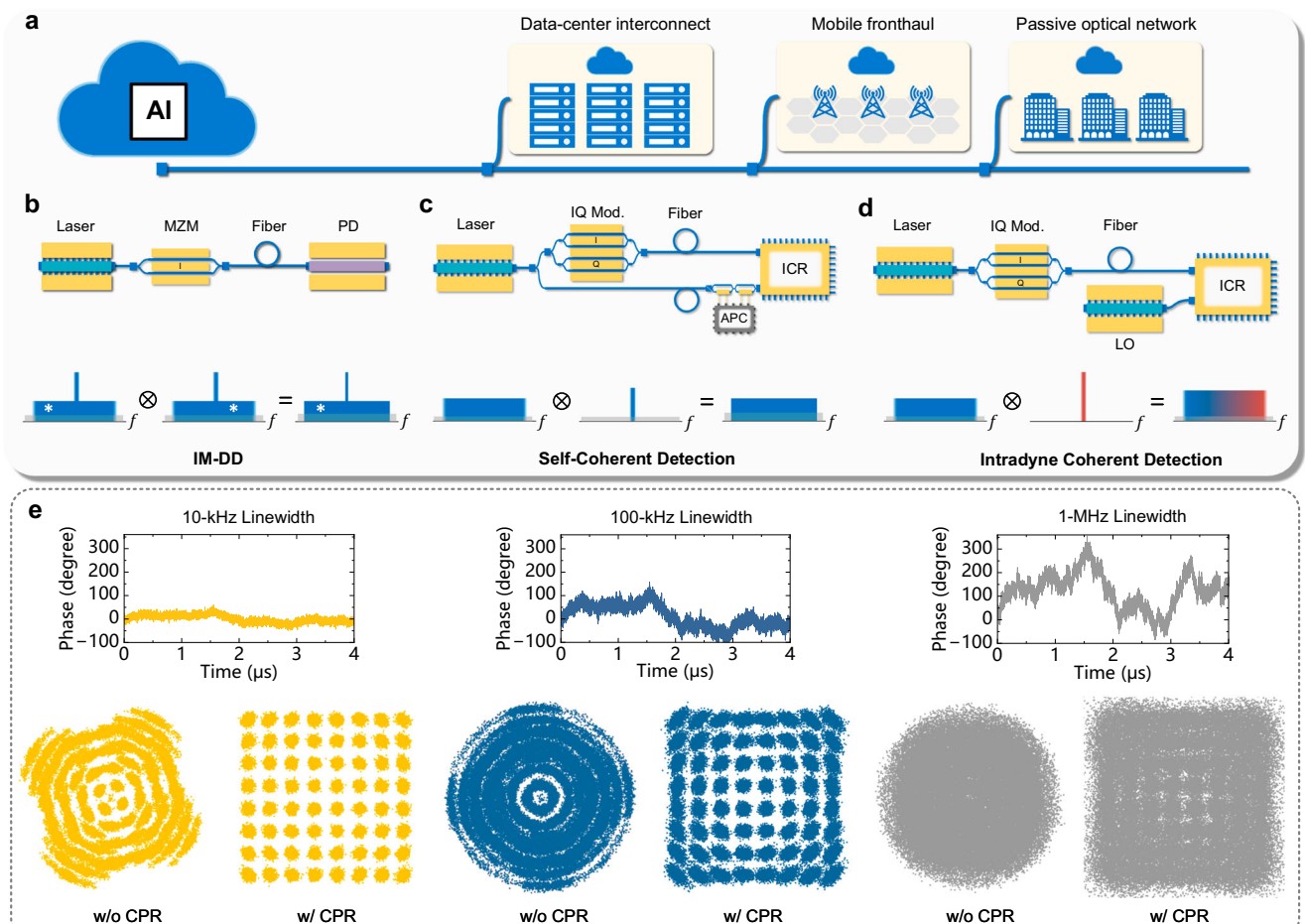

**Fig. 1 | Conventional optical transmission structures and the phase noise challenge for low-cost lasers in coherent detection. a** Short-reach optical transmission applications, encompassing data-center interconnects, mobile fronthaul, and passive optical networks. AI artificial intelligence. **b** The architecture of intensity-modulation with direct detection (IM-DD) scheme. DFB laser distributed feedback laser, MZM Mach-Zehnder modulator, PD photodiode. **c** The architecture of self-coherent schemes. IQ Mod. in-phase/quadrature (IQ) modulator, APC automatic polarization controller, ICR integrated coherent receiver. **d** The architecture of the intradyne coherent detection scheme. ECL external cavity laser, LO local oscillator. **e** The numerically simulated phase fluctuation and constellations of the intradyne coherent detected signal with laser linewidth sum of 10 kHz, 100 kHz, and 1 MHz. CPR carrier phase recovery, w/o without, w/ with. Here the CPR refers to the conventional time-domain pilot enabled phase recovery. The simulation is based on a 45-Gbaud 64-ary quadrature amplitude modulation (64-QAM) signal.

To extend the spectral efficiency and distance, self-coherent schemes are also potential candidates[14,15]. As illustrated in Fig. 1c, an optical carrier co-propagates with the information-bearing signal through an additional dimension, such as the spatial dimension in self-homodyne architecture. The optical carrier serves as a remote LO for signal recovery. With accurate optical path matching between the fiber pair[16], the laser phase noise can be mostly canceled. Nevertheless, remote LO halves the average spectral efficiency per fiber, and the randomly varying polarization state of the remote LO requires sophisticated endless polarization controllers[15] or complementary branches[17] in the receiver structure.

Moreover, the intradyne coherent receiver in Fig. 1d is widely deployed in long-haul and metro networks. It has the merit of superior receiver sensitivity, 4-dimensional modulation, and efficient digital compensation of channel impairments. Equipped with the high-bandwidth in-phase/quadrature (IQ) modulator[18], capacity-approaching probabilistic shaping (PS) technique[19], and precise in-phase/quadrature imbalance correction[20], state-of-the-art coherent systems have stridden over 2 Tb/s data rate in a single channel[21], or surpassed the spectral efficiency of 20 bit/s/Hz[22] leveraging kHz-level narrow linewidth lasers. However, when migrating to the cost-sensitive scenarios, distributed feedback (DFB) lasers with several MHz linewidth are preferred, making it difficult to recover high-order modulation formats.

The laser phase noise is one of the foremost impairments in coherent receivers, which stems from the unpredictable phase fluctuations between signal and LO lasers. As depicted in Fig. 1e, such phase noise causes a random and time-varying rotation on the signal constellation, where a larger linewidth corresponds to greater phase ambiguity. To combat this problem, commercial coherent optical transponders use carrier phase recovery (CPR) in the digital domain. This technique estimates the phase noise by transmitting receiver-known symbols as pilot symbols, and then inversely rotates the neighboring payloads[23]. Notably, the frame redundancy inevitably leads to the sacrifice of spectral efficiency. More importantly, with MHz-class linewidth lasers, the relative phase fluctuates rapidly over time, making it no longer a reliable estimation of adjacent symbols. Alternatively, a radio-frequency (RF) pilot tone-based phase recovery technique[24,25] is proposed, but it consumes precious quantization bits of DACs, enhancing the quantization noise and degrading signal SNR. Although an optical generation of pilot tone has been demonstrated with OFDM modulation[26], OFDM is much more susceptible to phase noise due to its longer symbol duration. Besides, the potential broad application of optical pilot tone in modern optical communication remains unexplored. Furthermore, overcoming phase noise in high-order modulation standard coherent systems with low-cost lasers has not yet been demonstrated.

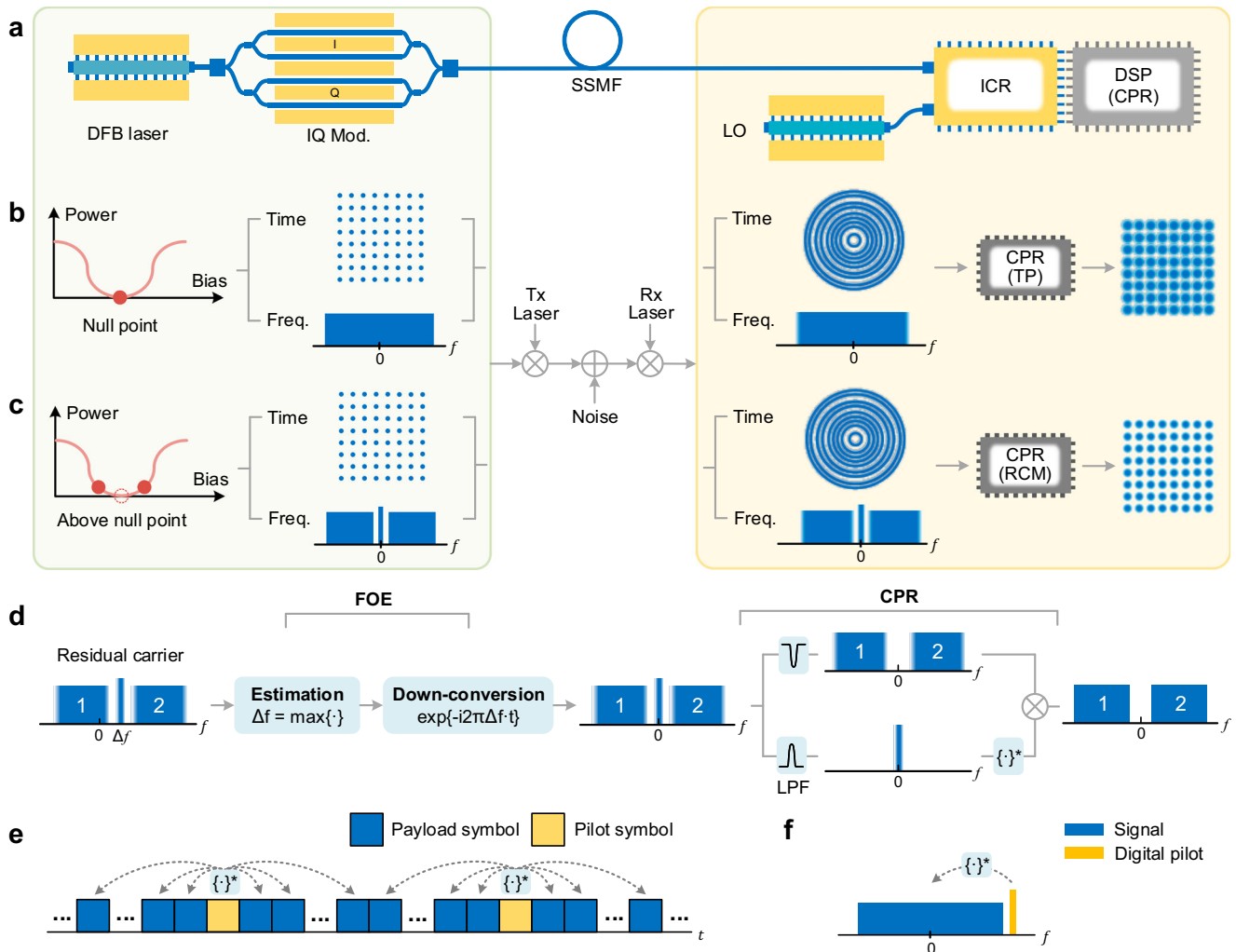

**Fig. 2 | Concept of the proposed residual carrier modulation (RCM) scheme.**
**a** The architecture of low-cost standard coherent transceivers using distributed feedback (DFB) laser at the transmitter and receiver. IQ Mod. in-phase/quadrature modulator, SSMF standard single-mode fiber, LO local oscillator, ICR integrated coherent receiver. The comparison of **b** conventional time-domain pilot (TP) and **c** the proposed RCM enabled carrier phase recovery (CPR). The constellation diagrams and optical spectra at the transmitter, the received electrical spectra, and constellation diagrams after phase recovery are shown. The direct current (DC) bias of modulators is plotted on the left of the transmitter spectra. Freq. frequency, Tx transmitter, Rx receiver. **d** The digital signal processing procedure for the residual carrier modulation to recover the phase. $\Delta f$ is the estimated frequency offset value. $\{\cdot\}^*$ denotes the complex conjugate operation. FOE frequency offset estimation, LPF low-pass filter. **e** The frame structure of the time-domain pilot-based CPR. **f** The principle of frequency-domain pilot tone-based CPR.

In this work, we present a residual carrier modulation (RCM) technique integrated with subcarrier multiplexing to achieve high-speed and high spectral efficiency in modern coherent optical communication systems using low-cost, large linewidth lasers. Our quantitative evaluation demonstrates the advantage of our phase tracking method over conventional time-domain and frequency-domain pilot techniques. We also showcase its potential in diverse scenarios such as data-center interconnects and fiber-wireless access networks. By slightly adjusting the bias point of the IQ modulator, we introduce the residual carrier that originates from the identical laser source as the signal. This approach enhances the tracking ability of phase noise by facilitating the beating between the signal and residual carrier in the digital domain. For the data-center interconnect scenario, we achieve net single-channel 1.0-Tb/s PS-256-QAM signal transmission over 80-km standard single-mode fiber (SSMF) using a 3-MHz linewidth laser. For high-capacity transmission, the residual carrier introduces negligible optical signal-to-noise ratio (OSNR) reduction and is well-suited for 16 × 900-Gb/s wavelength-division-multiplexed (WDM) data-center interconnects using transceiver laser with both 1-MHz linewidths. For the fiber-wireless access network scenario, we report high-fidelity analog radio-over-fiber fronthaul delivering 512-QAM signals, leading to a >2-Tb/s common public radio interface (CPRI)-equivalent rate within only 18.85-GHz optical bandwidth. The proposed methodology integrates the concept of self-homodyne detection with the conventional intradyne coherent system, thus significantly relaxing the hardware requirement. This reshapes various short-reach optical communications in the age of artificial intelligence.

## Results

### Phase recovery using residual carrier modulation

In the standard intradyne coherent architecture (Fig. 2a), the utilization of a low-cost DFB laser can cause severe phase noise due to the rapid and independent phase fluctuations between the signal laser and LO. The conventional approach inserts time-domain pilot (TP) symbols periodically between the blocks of payload symbol to probe the phase rotation (Fig. 2e). However, for lasers with a large linewidth, the recovered constellation diagrams are still blurred as the TP struggles to track rapid phase changes (Fig. 2b). Figure 2c illustrates the concept of residual carrier modulation. At the transmitter, the IQ modulator operates at a bias point slightly deviated from the null point,

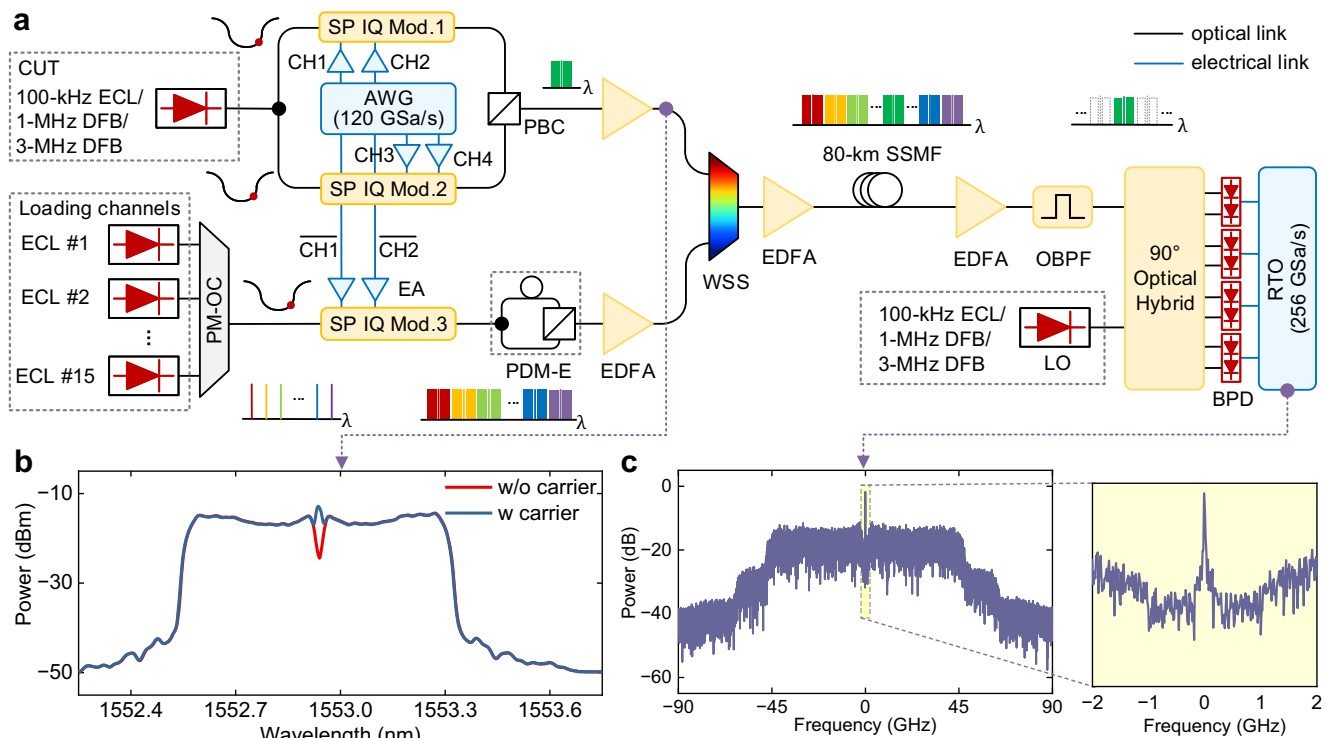

**Fig. 3 | Residual carrier modulation-enabled terabit coherent transmission system with high-order modulation formats. a** The experimental setup of 16-channel wavelength-division multiplexed (WDM) probabilistic-shaped 256-ary quadrature amplitude-modulation (PS-256-QAM) signals. CUT channel under test, ECL external cavity laser, DFB distributed feedback laser, CH channel, PM-OC polarization-maintaining optical coupler, SP IQ Mod. single-polarization in-phase/quadrature modulator, PBC polarization beam combiner, EA electrical amplifier, PDM-E polarization-division-multiplexing emulator, EDFA erbium-doped fiber amplifier, SSMF standard single-mode fiber, WSS wavelength-selective switch, OBPF optical band-pass filter, LO local oscillator, BPD balanced photodetector, RTO real-time oscilloscope. **b** The measured optical spectra of the channel under test with or without the residual carrier. w/ with, w/o without. **c** The received electrical spectrum after frequency offset removal.

generating a residual optical carrier for frequency and phase recovery. To cut off the crosstalk from the signal band, we propose the integration of dual-band subcarrier modulation (SCM)[27] to reserve a narrow guard band. After beating with the LO at the receiver, the residual carrier becomes the same blurred as the signal bands. Nonetheless, we leverage it for phase recovery. It should be noted that the frequency offset prevents the residual carrier from being blocked in the receiver analog circuit. As depicted in Fig. 2d, we initially remove the frequency offset by identifying the peak in the electrical spectrum and down-convert the signal and residual carrier to the baseband. A digital low-pass filter (LPF) is then employed to extract the residual carrier. The signal and residual carrier are subsequently beaten in the digital domain to simultaneously eliminate the two components of phase noise, as elaborated in Eq. (1).

$$\varphi'_S - \varphi'_{RC} = (\varphi_S - \varphi_{LO}) - (\varphi_{RC} - \varphi_{LO})$$
$$= \varphi_S - \varphi_{RC} = 0 \tag{1}$$

Here $\varphi'_S$ and $\varphi'_{RC}$ represent the phase of the digital signal and residual carrier after signal-LO beating. $\varphi_S$, $\varphi_{LO}$ and $\varphi_{RC}$ denote the phase of the optical signal, LO, and residual carrier, respectively. Similar to the concept of self-homodyne detection, the signal and the residual carrier originate from the same laser and traverse the same optical path. One essential difference is that the beating of the signal and residual carrier is accomplished in the digital domain, relaxing the hardware requirement. Additionally, residual carrier modulation eliminates the frame redundancy and offers modulation format transparent signal processing.

In contrast to the traditional pilot-based phase recovery method, our approach offers substantial improvements. The TP-based phase

recovery, involving the insertion of uniformly distributed time-domain pilot symbols at the transmitter as in Fig. 2e, aids in phase tracking at the receiver after channel equalization. However, it is not optimal from the point of the entire mathematical system model. In the coherent system, the transmitted signal first undergoes distortion due to fiber channels and is subsequently influenced by phase noise at the receiver. Therefore, in the absence of the mathematical commutative property, compensating for phase noise should precede the application of the channel equalizer to mitigate inter-symbol interference (ISI)[10]. An alternative phase recovery scheme involves utilizing a RF pilot tone[24,25], as depicted in Fig. 2f. This method introduces a frequency-domain RF tone in the digital signal processing (DSP) stage before the digital-to-analog converter (DAC) at the transmitter. Filtering the pilot tone at the receiver also allows for phase noise mitigation. Nevertheless, a drawback lies in the RF pilot tone's occupation of the precious effective number of bits (ENOB), thereby amplifying quantization noise.

## Experimental setup of low-cost coherent system

Figure 3 a illustrates a 16-channel WDM high-speed coherent transmission system with different laser linewidth configurations. For the channel under test (CUT), we employ a 100 kHz external cavity laser (ECL), 1-MHz DFB laser, or 3 MHz DFB laser as the signal laser source to evaluate transmission performance under varying phase noise conditions, respectively. A 120-GSa/s arbitrary waveform generator (AWG) with 45 GHz 3-dB bandwidth generates the eletrical 2 × 45-GBaud PS-256-QAM signal. We use two single-polarization IQ modulators to realize polarization-division-multiplexing (PDM). Both modulators are biased slightly above the null point to introduce the residual optical carrier. The transmitted optical spectrum of the CUT seeded by a 3 MHz linewidth DFB laser is depicted in Fig. 3b. For the loading

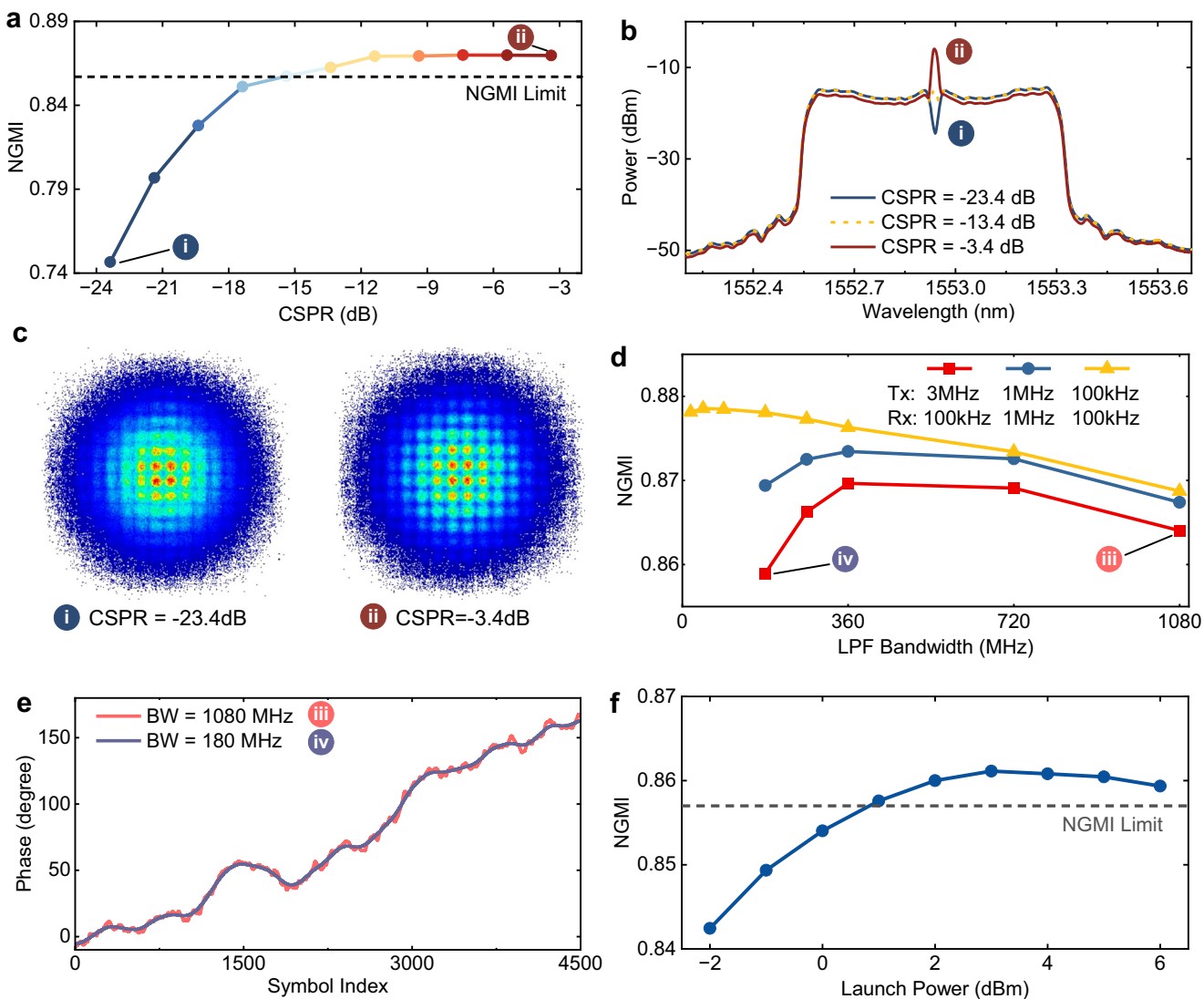

**Fig. 4 | Parameter optimization of residual carrier modulation in single-channel net 1.0-Tb/s PS-256-QAM transmission. a** The measured normalized generalized mutual information (NGMI) for different carrier-to-signal power ratios (CSPR) in the single-channel back-to-back (BTB) scenario. **b** The transmitted optical spectra with a CSPR of -23.4 dB, -13.4 dB and -3.4 dB, respectively. All the spectra are in 0.02 nm resolution. **c** The constellation diagrams with a CSPR of -23.4 dB and -3.4 dB. **d** The measured NGMI for different digital low-pass filter (LPF) bandwidths with different signal (Tx) and local oscillator (Rx) lasers with a CSPR of -11.4dB. The configuration includes 3 MHz Tx with 100 kHz Rx, 1 MHz Tx with 1 MHz Rx, and 100 kHz Tx with 100 kHz Rx. Tx, transmitter; Rx, receiver. **e** The recovered phase for different symbol indexes with a low-pass filter bandwidth (BW) of 180 MHz and 1080 MHz for 3 MHz Tx and 100 kHz Rx lasers. **f** The measured NGMI for different launch powers in a single-channel 80 km transmission scenario with 3 MHz Tx and 100 kHz Rx lasers with a CSPR of -11.4dB.

channels, we combine 15 external cavity lasers with 125 GHz spacing using a polarization-maintaining optical coupler. Then, they are fed into another IQ modulator and passed through a split-and-decorrelation-based emulator. The CUT and loading channels are boosted by two erbium-doped fiber amplifiers (EDFA), respectively. We set a programmable wavelength-selective switch to combine and flatten the wavelength channels simultaneously. The WDM super-channel is transmitted through up to 80-km standard single-mode fiber (SSMF).

At the receiver, we use another laser with a linewidth ranging from 100 kHz−3 MHz as the LO. An optical band-pass filter (OBPF) selects the desired channel. After intradyne coherent detection, the 256 GSa/s real-time oscilloscope (RTO) captures the electrical waveforms for offline digital signal processing. Figure 3c shows the received electrical spectrum after frequency offset removal. For comparative analysis, we also implement the conventional time-domain pilot-based phase recovery method. An equivalent net bitrate is maintained by adopting

a single-band 90-Gbaud signal with the same modulation format. Detailed descriptions of the digital signal processing stacks are provided in the Methods section.

### RCM-enabled 1-Tb/s single-channel transmission with a 3-MHz DFB laser

In this study, we assess the transmission penalty using the generalized mutual information (GMI), a key metric representing the maximum achievable data throughput for a bit-wise decoder[28]. Notably, the GMI is dependent on the modulation format. After normalization, the normalized GMI (NGMI) becomes a modulation-independent metric for predicting transmission performance after the forward-error correction (FEC), with an abundant NGMI threshold. Therefore, we also use the NGMI to provide the error-free net data rate after FEC. Here, the NGMI threshold is set at 0.857 with a corresponding code rate of 0.826[29]. In the single-channel scenario, we choose a source entropy (maximum achievable GMI) of 13.92 bits per four-dimensional

(4-D) symbol (6.96 bits in each polarization) for evaluation. For WDM transmission, a source entropy of 12.82 bits/4D-symbol is selected.

A high-quality residual carrier is paramount to achieve optimal digital beating of residual carrier and signal for accurate phase noise cancellation. This entails maintaining an appropriate carrier-to-signal power ratio (CSPR) and ensuring a suitable filter bandwidth to preserve the integrity of the residual carrier. We first optimize the parameters of RCM in the single-channel back-to-back scenario. We choose a digital Gaussian filter to extract the residual carrier. As shown in Fig. 4a, a CSPR of -11.4 dB is sufficient, ensuring that the phase information remains untainted by noise or interference. Since the power of the residual carrier is much smaller than the signal, the reduction in the effective optical signal-to-noise ratio is less than -0.3 dB (see Supplementary Note 5), which can be ignored. The measured transmitted optical spectra are shown in Fig. 4b. Subsequently, we delve into an assessment of the low-pass filter (LPF) bandwidth for the extraction of the residual carrier. The measured NGMI for different 3 dB bandwidths of LPF is illustrated in Fig. 4d, considering different laser configurations. To streamline the statement, if not specified, the received LO laser is a 100 kHz ECL. The results highlight the necessity for an adequate LPF bandwidth tailored to the laser linewidth, ensuring the filtration of phase information without introducing excessive noise. As revealed in Fig. 4e, oversized bandwidth introduces high-frequency noise or signal bands that affect the phase information similarly to modulation. Conversely, undersized bandwidths yield overly smoothed results, causing the loss of precise phase change information. With these above optimized parameters, the signal is transmitted over an 80 km fiber link. Figure 4f demonstrates the measured NGMI for different launch powers in the single-channel 80 km SSMF transmission with 3 MHz signal and 100 kHz LO lasers. The optimal launch power is 3 dBm and we obtained an NGMI above the NGMI limit of 0.857. Therefore, the net data rate is 1.0 Tb/s (See methods for the net bitrate calculation of PS signals) considering the FEC overhead with a record laser linewidth sum and symbol duration product of $6.89 \times 10^{-5}$ (ref. [30]).

## Performance of RCM under different laser linewidth configurations

In this study, we conduct a comprehensive evaluation of the performance of RCM and conventional TP-enabled phase recovery. Firstly, we assess the performance of RCM and TP-enabled phase recovery in simulations where the sole impairment is phase noise. Figure 5a illustrates the phase noise-induced penalty in the required OSNR for the NGMI limit of 0.857 as a function of different linewidth sum and symbol duration product $\Delta f \times T_s$. The modulation format is PS-256-QAM with an entropy of 13.92 bits/4D-symbol. This result emphasizes a significant enhancement achieved by the RCM scheme. The RCM scheme surpasses the upper limit of the TP method and makes it possible for an application of low-cost lasers with linewidth exceeding megahertz. Specifically, for an OSNR penalty of 1 dB, the RCM scheme enhances the maximum tolerable $\Delta f \times T_s$ value by an order of magnitude compared to the TP method, which means the acceptable transceiver laser linewidth sum can be increased from 100 kHz to 1 MHz for the 45-Gbaud PS-256-QAM signal.

In our proof-of-concept single-channel experiments, the NGMI with different received OSNR values at the back-to-back scenario is measured, as shown in Fig. 5b. We observe that the combination of 1 MHz and 3 MHz linewidth signal lasers with a 100 kHz LO incurs OSNR penalties of 0.82 dB and 2.05 dB, respectively, compared to the 100 kHz linewidth ECL. These penalties are quite close to the simulation result in Fig. 5a, where the 1 MHz and 3 MHz lasers bring OSNR penalties of 0.5 dB and 1.25 dB, respectively. Besides, as the phase noise is related to the laser linewidth sum of signal and LO lasers, the exchange of signal and LO lasers will not bring OSNR penalty at the back-to-back case, as shown in the 3 MHz Tx laser or 3 MHz Rx laser curves in Fig. 5b.

Figure 5c demonstrates the phase recovery performance of the conventional TP scheme. Although the gross GMI will improve and saturate as the increase of pilot ratio, a significant gap is observed when substituting the signal laser from 100-kHz ECL to 3 MHz DFB. Moreover, regarding the net GMI excluding the pilot overhead, the penalty is even greater. Specifically, the net GMI penalty is 3.13 bits/4D-symbol. However, for the residual carrier modulation, this penalty is only 0.17 bits/4D-symbol. For the same data rate PS-256-QAM signal with the 3 MHz signal laser and 100 kHz LO, the RCM improves the net GMI from 8.39 bits/4D-symbol to 11.85 bits/4D-symbol compared to TP-based phase recovery, corresponding to 41.2% enhancement in the net bitrate. The recovered constellations with RCM and TP schemes are shown in Fig. 5e. A significant signal quality improvement can be obtained by the RCM, especially for the 3 MHz linewidth case. Figure 5f compares the phase recovery performance of the proposed RCM and RF pilot tone-based phase recovery. The measured NGMI is presented for the same data rate PS-256-QAM signal, using a 3 MHz signal laser and a 100 kHz LO at the back-to-back scenario with varying receiver-side pilot tone-to-signal power ratios (PTSPR). Receiver-side PTSPR denotes the measured power ratio between the pilot tone and signal in the electrical domain. As RF pilot tone scheme can also track phase continuously, at moderate pilot tone power levels (approximately −20 dB PTSPR), it exhibits comparable performance to RCM. However, as the RF pilot tone occupies the DAC quantization bits, further increasing its power leads to enhanced quantization noise, thereby degrading NGMI performance. In contrast, RCM scheme can further improve performance by increasing the PTSPR. The peak GMI performance difference between the two schemes is 0.45 bits/4D-symbol (See Supplementary Note 7). Notably, our experiment utilized DACs with 8 bits vertical resolution and an effective number of bits (ENOB) of about 5.5 bits (Keysight 8194A). In practical scenarios with lower DAC resolution, the advantage of RCM would be even more pronounced.

To evaluate the performance of RCM after transmission, we show the measured GMI of different transceiver laser configurations after single-channel 80 km fiber transmission in Fig. 5d. For a fair comparison, the OSNR values of the transmitted signal are fixed at 40 dB in all the laser configurations. It can be observed that the exchange of the signal laser and LO with different linewidths, while maintaining the same laser linewidth sum and symbol duration product, will not introduce obvious performance variation. The slight degradation when the linewidth of the LO laser is larger is attributed to the equalization-enhanced phase noise (EEPN)[31].

## The 16-channel WDM Transmission using low-cost DFB transceiver lasers

Traditional self-coherent schemes typically necessitate a high CSPR, commonly around 10 dB[32–34], to mitigate the signal-signal beating interference (SSBI) as a perturbation. However, this prerequisite imposes limitations on the number of signal channels and, consequently, the overall system capacity in a WDM transmission due to the restricted output power of EDFAs[35]. In contrast, the RCM method only requires a typical CSPR value below -10 dB, thereby addressing this limitation. In this study, we assess the WDM transmission performance of RCM to reveal its advantages. The transmitted signal is PS-256-QAM with an entropy of 12.82 bits/4D-symbol. Figure 6a shows the measured NGMI of the 9th channel as a function of the launch power per channel, employing signal and LO lasers of 100-kHz ECL or 1-MHz DFB. The GMI reduction associated with 1 MHz transceiver lasers, as compared to 100-kHz transceivers, is merely 0.065 bits per 4D-symbol. The optimal launch power is identified as 1 dBm per channel after an 80 km 16-channel WDM transmission. This analysis reveals a trade-off between the SNR and fiber nonlinearity. In comparison to single-channel transmission with 1 MHz transceiver lasers, the WDM setup introduces a GMI degradation of only 0.352 bits/4D-symbol

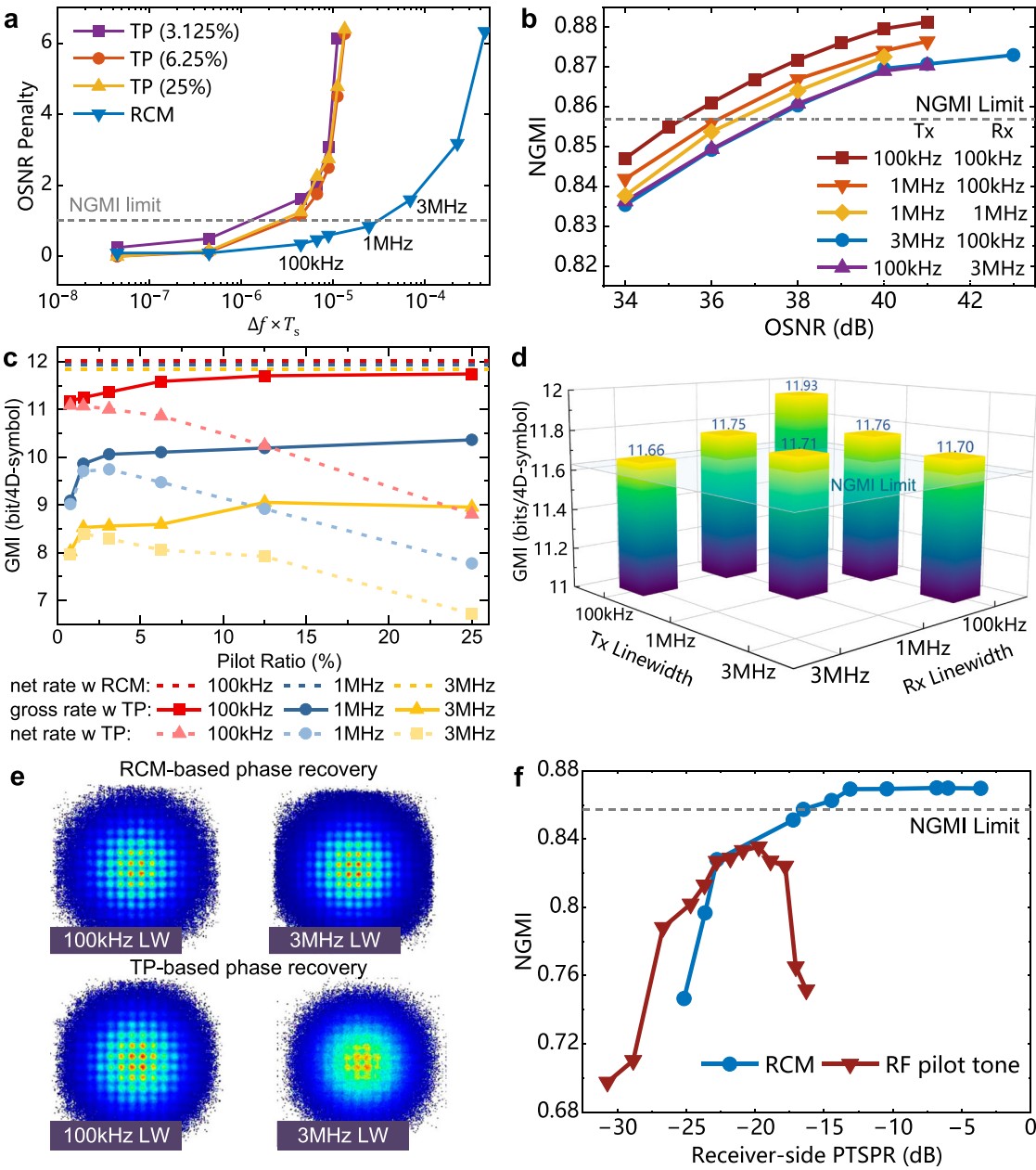

**Fig. 5 | Simulation and experimental results of the single-channel probabilistic-shaped 256-ary quadrature amplitude-modulation (PS-256-QAM) signal under different transceiver laser setup. a** The optical signal-to-noise ratio (OSNR) penalty for different linewidth sum and symbol period product $\Delta f \times T_s$ with conventional time-domain pilot (TP) enabled phase recovery and residual carrier modulation (RCM) under simulation. **b** The measured normalized generalized mutual information (NGMI) versus the OSNR for different signal (Tx) and local oscillator (Rx) lasers using RCM in the back-to-back experiment. Tx, transmitter; Rx, receiver. **c** The measured generalized mutual information (GMI) for the different pilot ratios of the TP-based phase recovery with 100 kHz, 1 MHz, and 3 MHz linewidth signal lasers in the back-to-back experiment at an OSNR of 41 dB. The results

of RCM are also plotted for comparison. The gross rate and net rate respectively denote the GMI that includes pilot overhead and the GMI that excludes pilot overhead. w, with. **d** The measured GMI for different signal and local oscillator lasers with RCM at a transmitted OSNR of 40 dB in the 80 km transmission experiment. **e** The recovered PS-256-QAM constellation for 100 kHz and 3 MHz signal laser with 100 kHz local oscillator using RCM scheme or TP-based phase recovery at an OSNR of 41 dB. LW linewidth, 4D 4-dimensional. **f** The measured NGMI using RCM or radio-frequency (RF) pilot tone for phase recovery under different receiver-side pilot tone to signal power ratio (PTSPR) with 3 MHz signal laser and 100 kHz local oscillator laser. The receiver-side PTSPR denotes the measured power ratio between the pilot tone and signal in the electrical domain.

(See Supplementary Note 6). The recovered constellations at the optimal launch power are shown in Fig. 6b.

Figure 6c illustrates the transmitted optical spectrum of the WDM system. The WDM channels are labeled from channel (CH) 1 to CH 16, arranged from low to high frequency. With RCM-based phase recovery, the 16 channels achieve an average NGMI of 0.90, as depicted in Fig. 6d. All channels are well above the NGMI limit of 0.857 after 80-km SSMF transmission. The net bitrate is 903.2 Gb/s/λ and the aggregated

capacity adds up to 14.45 Tb/s ($16\lambda \times 903.2$ Gb/s/λ). The results indicate the feasibility of RCM for low-cost and high-capacity data-center interconnects exceeding 10 Tb/s.

## RCM-enabled high-fidelity fiber-wireless access network with 512-QAM signal

Apart from data-center interconnects, fiber-wireless access networks become increasingly popular in short-reach scenarios. In such

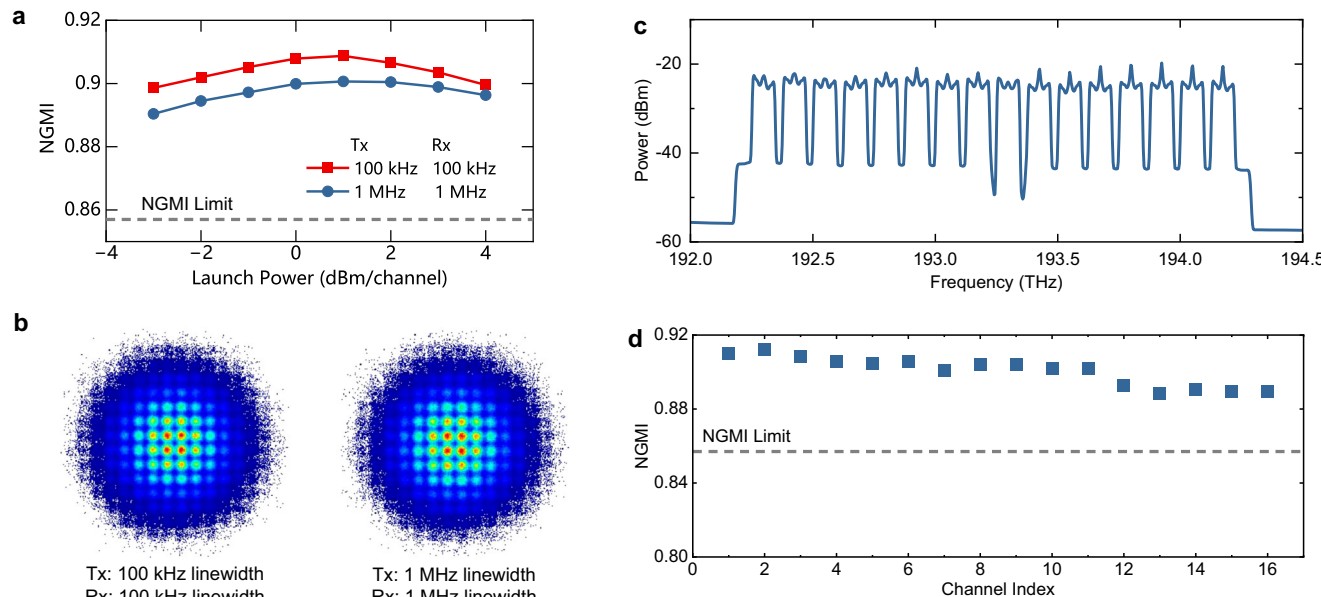

**Fig. 6 | Experimental results of $16\lambda \times 903.2$-Gb/s/$\lambda$ wavelength-division-multiplexed (WDM) transmission enabled by residual carrier modulation.** The modulation format is probabilistic-shaped 256-ary quadrature amplitude-modulation (PS-256-QAM) with an entropy of 12.82 bits/4D-symbol. **a** The measured normalized generalized mutual information (NGMI) as a function of different launch power per channel at the 9th channel. The linewidth of signal and lasers are both 100 kHz or 1 MHz. Tx transmitter, Rx receiver. **b** The recovered constellations at the optimum launch power for transceiver lasers with different linewidths. **c** The measured optical spectrum of the transmitted 16-channel WDM signal with a 0.02 nm resolution. **d** The measured NGMI for all 16 channels. The results are well above the NGMI limit of 0.857.

networks, fronthaul plays a crucial role in seamlessly connecting mobile and optical communications. The wireless waveforms are distributed from the centralized unit (CU) or distributed unit (DU) to the radio units (RUs) through optical fibers, as depicted in Fig. 7a. The analog radio-over-fiber (RoF) technique, which directly modulates the original wireless signal onto the optical carrier, is promising thanks to its high spectral efficiency and simplicity. As wireless communications are moving towards 1024-QAM, the transceiver distortion and laser phase noise in fronthaul hinder the adoption of high-order modulation format beyond the 64-QAM.

In this study, we demonstrate the application of the RCM technique for mitigating phase noise in analog RoF for fronthaul. As illustrated in Fig. 7a and b, we separate the left sideband (LSB) and right sideband (RSB) for the downlink and uplink, respectively. Such configuration not only separates the transmitter IQ imbalance-induced crosstalk, but also avoids the backward scattering for bi-directional transmission. Therefore, the SNR bottleneck is considerably alleviated for the analog RoF transmission of high-order modulation formats. The residual carrier, when beaten with the LO, serves for phase recovery, as depicted in the electrical spectra in Fig. 7c. With these impairments mitigated, the SNR can reach 30.8 dB at symbol rate of 17 Gbaud, satisfying the 29.1 dB SNR requirement of the 256-QAM format, as shown in Fig. 7d. This corresponds to an over 2 Tb/s/$\lambda$ CPRI-equivalent rate[36,37] using only an 18.85 GHz optical bandwidth. The SNR gradually degrades with symbol rate, attributed to increased in-band noise and bandwidth limitation. Figure 7e demonstrates the bit-error rate (BER) versus the modulation formats from 128-QAM to 512-QAM at 17 Gbaud. Even for the 512-QAM signal, the BER is well below $1 \times 10^{-2}$. The receiver sensitivity is evaluated to be around -18 dBm at the 15.3% open forward error correction (O-FEC) threshold of $1.8 \times 10^{-2}$ for the 512-QAM format, as shown in Fig. 7f. Finally, we assess a 12-channel WDM transmission over 10 km SSMF in Fig. 7g. The BER values of all channels in both the downlink and uplink are well below the O-FEC threshold. Therefore, we successfully demonstrate the transmission of 25.1-Tb/s ($12\lambda \times 2.089$ Tb/s/$\lambda$) aggregated CPRI-equivalent rate and 512-

QAM signals over 10-km SSMF. Besides, the linewidth is 100 kHz for both signal and LO lasers, corresponding to a record linewidth sum and symbol duration product of $1.18 \times 10^{-5}$ for 512-QAM format. By comparing the recovered constellations, the residual carrier modulation-based recovery effectively eliminates the phase blur, particularly for high-order formats, showcasing superior performance compared to the time-domain pilot-based scheme.

## Discussion

In summary, our study introduces a pioneering approach to recover the carrier phase by utilizing the residual carrier for beating the carrier and signal in the digital domain, in contrast to the optical beating used in self-coherent systems. This diverges fundamentally from traditional phase recovery methods employed in coherent optical communication systems. The deliberately preserved optical carrier serves as a crucial link connecting the signal phase with the carrier phase contaminated by the phase noise. This innovation enables the utilization of low-cost DFB lasers for conventional intradyne coherent transceivers. The elimination of external cavity lasers has the potential to significantly reduce costs by orders of magnitude.

Compared to conventional time-domain pilot-aided carrier phase recovery, RCM enhances the tolerable product of linewidth sum and symbol duration ($\Delta f \times T_s$) by more than an order of magnitude[21,22,38–41], which facilitates the deployment of low-cost DFB lasers with linewidth exceeding MHz for short-reach coherent transceivers. In the proof-of-concept experiments, we demonstrate the net 1.0 Tb/s/$\lambda$ 256-QAM signal transmission with a 3 MHz DFB laser using the RCM scheme, showing a 41% bitrate improvement compared to the time-domain pilot for phase recovery. As shown in Fig. 8a, we have set a record for the product of linewidth sum and symbol duration $\Delta f \times T_s$ with a net bitrate beyond 1 Tb/s.

Compared to conventional self-coherent architecture, RCM can significantly reduce the operating carrier-to-signal power ratio by around 20 dB. Therefore, the bottleneck of restricted channels for WDM transmission in self-coherent systems no longer exists. We

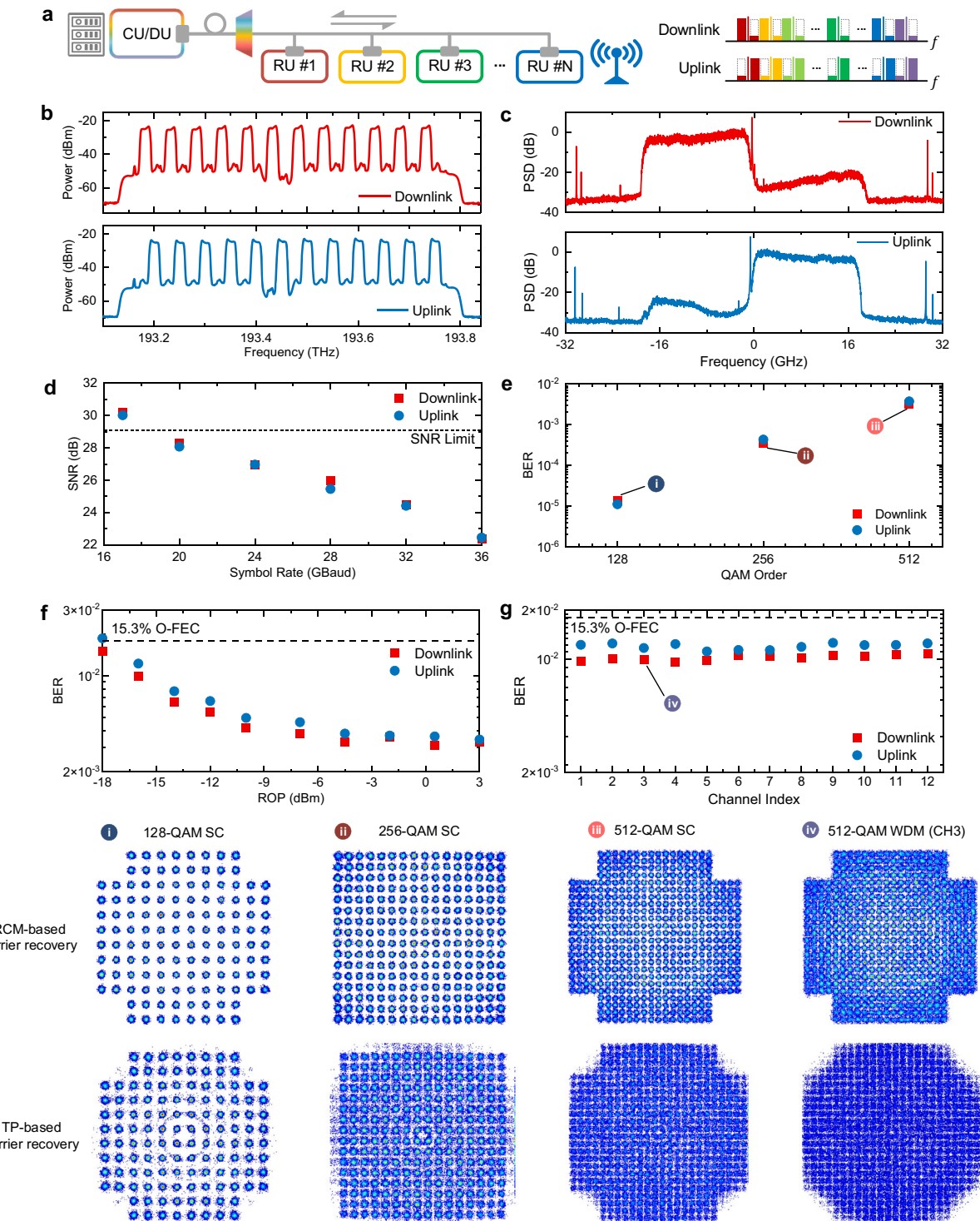

**Fig. 7 | Experimental demonstration of residual carrier modulation (RCM) for high-fidelity 512-ary quadrature amplitude-modulation (512-QAM) fiber-wireless access networks. a** The conceptional illustration of the fronthual connecting the centralized unit (CU) or distributed unit (DU) to the radio units (RUs). The proposed residual carrier-based bi-directional transmission with separate sidebands is illustrated on the right. **b** The measured optical spectra of the 12-channel 17-Gbaud 512-QAM signal for the downlink and uplink with the left sideband and right sideband. The channels are labeled from CH1 to CH12, arranged from low frequency to high frequency. CH, channel. **c** The received electrical spectra of the central 6th channel for the downlink and uplink. **d** The recovered signal-to-noise ratio (SNR) versus the symbol rate with 512-QAM at a single channel back-to-back scenario. **e** The measured bit-error rate (BER) for different modulation formats from 128-QAM to 512-QAM for single-channel 17-Gbaud signal at back-to-back. **f** The measured BER versus received optical power (ROP) with 17-Gbaud 512-QAM signal at single channel back-to-back scenario. **g** The measured BER for all the 12 WDM channels after 10 km transmission with 17-Gbaud 512-QAM. The insets show the corresponding recovered constellations with residual carrier modulation or time-domain pilot-based (TP) carrier recovery. SC, single channel.

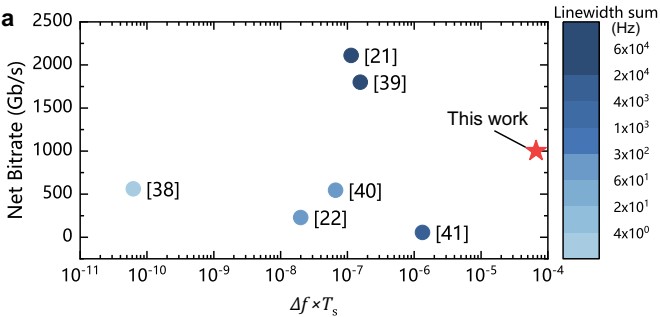
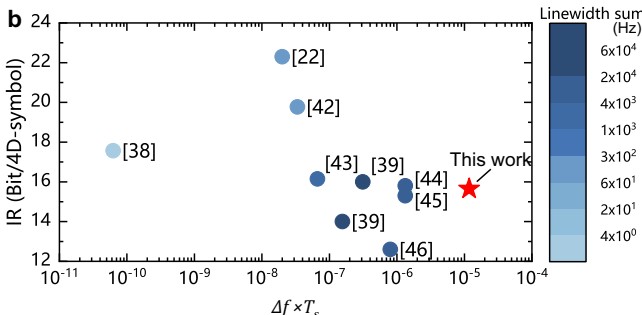

**Fig. 8 | State-of-the-art results. a** The net bitrate versus the product of laser linewidth sum and symbol duration $\Delta f \times T_s$. In this work, we achieve net 1.0-Tb/s PS-256-QAM single-channel transmission with a 3-MHz linewidth laser. **b** The information rate (IR) per 4D symbol versus the product of laser linewidth sum and symbol duration $\Delta f \times T_s$. In this work, we report high-order 512-QAM transmission using transceiver lasers with both 100 kHz linewidths. 4D 4-dimensional.

demonstrate $16\lambda \times 903.2$-Gb/s PS-256-QAM signal transmission over 80 km SSMF using a 1 MHz DFB laser pair as the transceiver laser source, showing only mild performance degradation compared to single-channel transmission.

To accommodate the phase noise challenge in analog radio-over-fiber systems, RCM provides an efficient solution to increase the SNR ceiling in conventional intradyne coherent systems[22,38,39,42–46]. We demonstrate high-fidelity analog RoF signal transmission with a 512-QAM format, which is also based on a record linewidth sum and symbol duration of $1.18 \times 10^{-5}$, as depicted in Fig. 8b.

The proposed RCM demonstrates a low-cost short-reach solution with minimal hardware complexity, uncompromised effective optical signal-to-noise ratio, and precise modulation-format-transparent phase tracking performance. Beyond its application in optical communication, the concept of residual carrier modulation may find its potential application in various areas such as precision measurement, continuous variable quantum key distribution, frequency-modulated continuous wave Lidar, and other sensing systems.

## Methods

### Detailed experimental setup and DSP stacks for PS-256-QAM transmission

At the transmitter, the WDM channels are composed of 15 ECLs for loading channels and one laser for the channel under test (CUT). The channel spacing is kept at 125 GHz. As the wavelength of the 1 MHz and 3 MHz DFB lasers is not tunable, we assess the performance of all the WDM channels by periodically changing the wavelength of the loading channels, which alternates the relative position of the CUT to the whole WDM spectrum. The linewidth measurement result is provided in Supplementary Note 1. The baseband PS-256-QAM signal with two subcarriers (45 GBaud × 2) is generated by an arbitrary waveform generator (AWG), with a 3 dB bandwidth of 45 GHz and a sampling rate of 120 GSa/s. All of the three IQ modulators have a 3-dB bandwidth of around 27 GHz. The modulators are biased slightly deviating from the null point (See Supplementary Note 10 for details on the bias control method). Then for the CUT, the signal on two polarizations is individually modulated and combined through the polarization beam combiner (PBC). The loading channels are jointly modulated with another modulator with the conjugate output of the AWG to generate different signals. It was evaluated both in a back-to-back configuration and with 80-km standard single-mode fiber.

At the receiver, the signal after the optical band-pass filter is sent to the optical 90° hybrid followed by balanced photodiodes (BPDs) with 70 GHz 3 dB bandwidth. It is subsequently sampled by a 256 GSa/s real-time oscilloscope with 59-GHz bandwidth. A detailed single-channel experimental setup is also depicted in Supplementary Note 2, Fig. S2a.

The DSP stack is shown in Supplementary Note 2, Fig. S2b. At the transmitter, the data is mapped to 40960 PS-256-QAM symbols first,

which is framed by a 4352-symbol preamble for synchronization and channel equalization. After up-sampling, the signal is pulse-shaped with a roll-off factor of 0.05. Then the signal is re-sampled to match the sampling rate of AWG. Subsequently, a dual-band subcarrier modulation is conducted with an optimized guard band interval of 2.0 GHz. A linear pre-emphasis is applied to compensate for the transmitter bandwidth limitation.

At the receiver, the captured signal is first re-sampled to 4 samples-per-symbol (SPS). Then a chromatic dispersion compensation (CDC) is employed for 80 km SSMF transmission. Subsequently, the frequency offset estimation (FOE) and carrier phase recovery are completed with the aid of residual carrier, as shown in Supplementary Note 2, Fig. S2c. After frame synchronization, the channel equalization is performed by 3rd-order Volterra nonlinear equalizer (VNLE). The 1st, 2nd, and 3rd-order memory length of the VNLE is optimized to 161, 41, and 11, respectively. After subcarrier demultiplexing and down-sampling to 1 SPS, we use the blind phase search (BPS) algorithm[47] to finely correct the residual phase fluctuation within ± 5. 3°. Then the GMI or NGMI is calculated as the performance metric.

### Calculation of net bitrate with the PS signal

Assume a concatenated FEC with a total code rate $R_c$ of 0.826 is used, the threshold of NGMI is 0.857. Then the net bitrate (NBR) of PS-$M$-QAM per subcarrier in dual polarization can be calculated as

$$\text{NBR} = 2 \times \left[ H - (1 - R_c) \log_2 M \right] \times B. \tag{2}$$

Here $H$ is the source entropy for 2-dimensional symbols and $B$ is the baud rate. For 45-Gbuad dual-subcarrier PS-256-QAM signal with a source entropy of 13.92 bits/4D-symbol. The net bit rate is 1.0 Tb/s ($= 2 \times 2 \times [13.92/2 - (1 - 0.826) \times \log_2 256] \times 45\,\text{Gb/s}$). For the WDM channels with an entropy of 12.82 bits/4D-symbol, the net bit rate is 903.2 Gb/s ($= 2 \times 2 \times [12.82/2 - (1 - 0.826) \times \log_2 256] \times 45\,\text{Gb/s}$).

### The detailed experimental setup and DSP for front-haul transmission

Figure S4 in Supplementary Note 4 provides a detailed illustration of the experimental setup. At the transmitter, we use a 1550 nm external cavity laser (ECL) with 100 kHz linewidth as an optical source of the channel under test (CUT). The 17-Gbaud LSB/RSB 512-QAM signal is generated by a 120-GSa/s AWG and then modulated through a 27 GHz 3 dB bandwidth IQ modulator. The bias is deviated from the null point to introduce a residual carrier with - -15 dB carrier-to-signal power ratio. For the loading channels, 11 ECLs spacing at 50 GHz are combined and fed into IQ modulator 2 simultaneously. The CUT and loading channels are respectively amplified, polarization division multiplexed and combined using a wave-shaper. After 10 km SSMF transmission, an optical band-pass filter is placed to select the desired

wavelength. The demultiplexed signal is intradyne coherent detected with a 100-kHz local oscillator. After four 70 GHz BPDs detection, the electrical waveform is captured by a 128-GSa/s real-time oscilloscope for offline DSP.

In the transmitter-side DSP, 32768 512-QAM symbols are mapped from binary bits, which is framed by a 3072-symbol preamble for synchronization and channel equalization. After up-sampling, root-raised cosine shaping is conducted with a roll-off of 0.1. Then the sequence is digitally up-converted for LSB and RSB modulation, in which a 1 GHz guard band is reserved for the residual carrier. After re-sampling to AWG sampling rate, linear pre-emphasis is applied to improve the bandwidth response. In the receiver-side DSP, we first emulate a DC block by subtracting the average value from the photocurrents of the in-phase/quadrature on X and Y polarizations. Then the detected waveform is 3 times re-sampled, frequency and phase recovered by the residual carrier, synchronized, and equalized based on the preamble. The multi-input multi-output (MIMO) equalizer has 3-order sparse Volterra kernels with lengths of (81, 11, 11). We use the blind phase search algorithm to finely correct the residual phase noise within ±8°.

## Data availability
The data that support the plots within this paper and other findings of this study are available on Zenodo database [https://doi.org/10.5281/zenodo.12513545]. All other data used in this study are available from the corresponding authors upon request.

## Code availability
The codes that support the findings of this study are available from the corresponding authors upon request.

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

## Acknowledgements

This work was supported by National Natural Science Foundation of China (62271010, F.Z.; U21A20454, F.Z.; 62001287, Y.Z.) and the major key project of Peng Cheng Laboratory (F.Z.). This work was also supported by High-performance Computing Platform of Peking University. We would like to thank Prof. Xiaopeng Xie and Dr. Chenbo Zhang from Peking University for providing the 3-MHz DFB laser and the helpful discussion. We also thank Prof. Dan Lu and Mr. Hao Song from Institute of Semiconductors, Chinese Academy of Sciences for their assistance with the laser linewidth measurement.

## Author contributions

X.F., Y.Z. and F.Z. conceived the concept of residual carrier modulation. X.F. and Y.Z. performed the experiment and analyzed the results. X.F., Y.Z. and F.Z. wrote the manuscript. X.C., W.H., Z.H. and S.Y. participated in preparing the manuscript and contributed to the discussions. All authors reviewed and revised the paper. F.Z. supervised the work.

## Competing interests

X.F. and F.Z. have filed a patent application on the residual carrier modulation method for laser phase noise compensation: CN202410809830.2, filed 21 June 2024. The remaining authors declare no competing interests.
