## [Peer Review File · Nature Communications]

Overcoming laser phase noise for low-cost coherent optical communicationREVIEWER COMMENTS

Reviewer #1 (Remarks to the Author):

This paper describes the use of residual carrier modulation (RCM) to reduce laser phase noise and employ low-cost lasers. By adjusting the bias point of the IQ modulator, the residual carrier can be effectively used to track the phase changes. The authors simulated and compared the performance for the PS-256-QAM signals with TP-based phase recovery and showed the superior performance of the RCM system. They also demonstrated the experimental results using a high-performance oscilloscope. It is a good way to utilize the residual carrier to minimize the cost and the system requirement. They achieved 1.0 Tb/s PS-256-QAM transmission with a 3 MHz linewidth laser, which is a very useful result with a simple, low-cost system. Although the reviewer wonders if the stability of the bias point is enough for long time operation, the manuscript seems to be well prepared and it could be published.

Reviewer #2 (Remarks to the Author):

This paper propose and demonstrate an simplified analog pilot tone carrier phase recovery method for coherent optical communication systems. This method direct uses offset MZM bias control to introduce a DC carrier pilot tone signal and therefore is simpler than previous RF pilot tone based method. The result shown in this paper is interesting and might be useful for the industry, but both novelty and technical depth is not significant enough to merit publication in Nature Communication due to following two reasons.

1) The method itself is a natural extension of RF pilot tone method. The reason to use RF pilot tone not DC carrier has practical reasons such as the receiver TIA typically use AC coupled design (for high-speed signal quality reason) so the DC carrier will be removed after receiver TIA.

2) The major claim in the this paper is that the DC carrier method can drastically improve the laser linewidth tolerance, but this claim is quite questionable and the reviewer believes the results shown in Fig. 5 is incorrect. Fundamentally, analog pilot tone based methods are vulnerable to signal modulation patterns while digital pilot tone is not. So Digital pilot tone should fundamentally performs better than analog pilot tone based methods (at least comparable), not otherwise as claimed in this paper. The reviewer suspect that non-optimal phase recovery algorithm is used in this paper: for digital pilot tone based method, the phase estimation accuracy can be greatly improved by using multi-stage phase recovery algorithms as demonstrated in many published papers

Reviewer #3 (Remarks to the Author):

The authors propose a pilot-tone-assisted phase-noise compensation scheme in digital coherent transmission, especially, for short-haul transmission systems. The author's claim is that the novelty of the authors' proposed technique is that it optically generates pilot tones by detuning modulator bias voltage. However, I have concerns about the novelty of this work. The technique of receiving high multi-level signals with a wide linewidth laser by compensating phase noise with pilot tones has been widely studied since the early days of digital coherent transmission. The modulator bias detuning technique has also been proposed for optical OFDM systems. Therefore, I do not find the novelty and originality in the author's proposed method enough for publication in Nature Communications. However, it may be noteworthy that the authors have refined this technique for modern coherent transmission systems and applied it to high-multi-level signal transmission using probabilistic constellation shaping, achieving notable performance in terms of net rate and laser linewidth. The authors should cite appropriate prior works regarding the

techniques utilized in the proposed configuration to further clarify the novelty. Below are my concerns and questions.

1. A method of generating a residual carrier by detuning the modulator bias voltage from the null point as a pilot tone for phase noise compensation has already been proposed in the field of optical OFDM in Ref [1]. Also, the phase noise compensation method using the pilot tone shown in Fig. 2d is almost the same as the method proposed in Ref. [2]. The authors should cite these references for the description of the proposed configuration and clarify the authors' originality with respect to these previous works.

2. I think the intrinsic comparison between the proposed method and the conventional methods (RF tone method and TP method) is insufficient. Detuning the bias voltage causes distortion in the signal constellation. According to the supplements, I guess that the authors apply a Volterra nonlinear equalizer to the receiver side to deal with this, but it is not stated clearly. The bias detuning penalty and how it is addressed should be explained for comparison with conventional RF pilot tone schemes. Are the benefits of the proposed scheme sufficient compared to the high cost and complexity of DSP due to the Volterra nonlinear equalizer?

3. In a typical digital coherent transmitter, an auto-bias control is used to tune the bias of the IQ modulator to the null point. How did the authors achieve a stable bias voltage detuned from the null point?

4. I guess that the residual carrier component power after polarization splitting in a polarization diversity receiver is fluctuating due to polarization rotation and polarization-mode dispersion in the transmission line. That is, CSRR will be fluctuating. This is caused by interference between the residual carrier components between orthogonal polarizations. How did the authors manage to address this issue?

References

1. S. L. Jansen, I. Morita, and H. Tanaka, "Experimental demonstration of 23.6-Gb/s OFDM with a colorless transmitter," *OptoElectronics and Communications Conference (OECC)*, 2007, PD1-5.
2. S. L. Jansen, I. Morita, T. C. W. Schenk, N. Takeda, and H. Tanaka, "Coherent Optical 25.8-Gb/s OFDM Transmission Over 4160-km SSMF," *J. Lightwave Technol.*, 26, 1, 6-15, 2008.

We appreciate the careful review by the reviewers and have modified the manuscript in accordance with their suggestions. Here, we present a point-by-point reply (in blue) to the reviewers' comments (in black), as well as the action taken (in red).

Response to Reviewer #1

Comment: This paper describes the use of residual carrier modulation (RCM) to reduce laser phase noise and employ low-cost lasers. By adjusting the bias point of the IQ modulator, the residual carrier can be effectively used to track the phase changes. The authors simulated and compared the performance for the PS-256-QAM signals with TP-based phase recovery and showed the superior performance of the RCM system. They also demonstrated the experimental results using a high-performance oscilloscope. It is a good way to utilize the residual carrier to minimize the cost and the system requirement. They achieved 1.0 Tb/s PS-256-QAM transmission with a 3 MHz linewidth laser, which is a very useful result with a simple, low-cost system. Although the reviewer wonders if the stability of the bias point is enough for long time operation, the manuscript seems to be well prepared and it could be published.

Reply: We would like to express our gratitude to the reviewer for the positive feedback.

Regarding the bias point stabilization, the bias point can be fixed using the auto-bias control technique. Specifically, the dither detection-based auto-bias control technique [R1] can achieve precise and long-term stable arbitrary bias point control. Therefore, this technique is feasible to maintain a stable bias voltage in residual carrier modulation systems.

[R1] X. Li et al., "Arbitrary Bias Point Control Technique for Optical IQ Modulator Based on Dither-Correlation Detection," *J. Lightw. Technol.*, vol. 36, no. 18, pp. 3824-3836, 2018.

We appreciate the careful review by the reviewers and have modified the manuscript in accordance with their suggestions. Here, we present a point-by-point reply (in blue) to the reviewers' comments (in black), as well as the action taken (in red).

Response to Reviewer #2

General Comment: This paper propose and demonstrate an simplified analog pilot tone carrier phase recovery method for coherent optical communication systems. This method direct uses offset MZM bias control to introduce a DC carrier pilot tone signal and therefore is simpler than previous RF pilot tone based method. The result shown in this paper is interesting and might be useful for the industry, but both novelty and technical depth is not significant enough to merit publication in Nature Communication due to following two reasons.

Reply: We thank the reviewer for the thorough reading and the insightful suggestions to improve our work. The residual carrier is generated in the optical domain to provide hardware-efficient, continuous, and accurate phase tracking while avoiding the drawbacks of the conventional RF pilot tone-based method (we compare and confirm this point in the revised manuscript). We have carefully studied your comments and made the point-by-point response as follows.

Comment 1: The method itself is a natural extension of RF pilot tone method. The reason to use RF pilot tone not DC carrier has practical reasons such as the receiver TIA typically use AC coupled design (for high-speed signal quality reason) so the DC carrier will be removed after receiver TIA.

Reply: We would like to thank the reviewer for raising the valuable concern on the practical implementation issue. We have taken this point into consideration in the following aspects.

- (1) We would like to clarify that the proposed RCM method is based on intradyne detection rather than homodyne detection, in which a small frequency offset (FO) between signal laser and LO is set in prior. In such a configuration, the zero-frequency component at the transmitter is not equal to the zero-frequency component at the receiver, so it will not be blocked by the AC-coupled design once the frequency offset is stable.
- (2) In the first step of the receiver-side digital signal processing, we have emulated a DC block by subtracting the average value from the photocurrents of the in-phase/quadrature on X and Y polarizations. This operation can partly emulate the AC-coupled receiver and ensure the feasibility in practical cases.

-
- (3) In practical applications, wavelength control methods such as temperature control of the lasers can be applied to realize this frequency offset with long-term stability. These methods are already deployed in commercial wavelength-division-multiplexing (WDM) systems.
- (4) Moreover, we have measured the frequency offset fluctuation every 30 minutes over an 8-hour period in our experiment system. We use a 1-MHz low-cost DFB laser for the signal laser and an ECL for LO. As shown in Fig. R1, the results indicate that the frequency offset typically fluctuates around several hundred megahertz, which helps the residual carrier avoid the influence of DC isolation.

Fig. R1. The recorded frequency offset (FO) versus time over an 8-hour measurement

Comment 2: The major claim in this paper is that the DC carrier method can drastically improve the laser linewidth tolerance, but this claim is quite questionable and the reviewer believes the results shown in Fig. 5 is incorrect. Fundamentally, analog pilot tone based methods are vulnerable to signal modulation patterns while digital pilot tone is not. So Digital pilot tone should fundamentally performs better than analog pilot tone based methods (at least comparable), not otherwise as claimed in this paper. The reviewer suspect that non-optimal phase recovery algorithm is used in this paper: for digital pilot tone based method, the phase estimation accuracy can be greatly improved by using multi-stage phase recovery algorithms as demonstrated in many published papers.

Reply: To avoid confusion, we first categorize the carrier phase recovery methods as:

- 1) residual carrier-based method (or DC carrier) introduced in the optical domain;
- 2) time-domain pilot symbol-based method;
- 3) frequency-domain pilot tone-based method (or RF pilot tone);

In our manuscript, Fig. 5(a) and Fig. 5(c) compare residual carrier modulation (RCM, Case I) with the time-domain pilot (TP) scheme-based phase recovery (Case II). RCM outperforms the TP scheme mainly owing to its continuous phase tracking, especially in MHz-linewidth cases. The performance of the digital pilot tone (DPT, Case III) that is also referred as the RF pilot tone scheme (to distinguish it from the optical domain

residual carrier), has been added in the revised manuscript.

Following the reviewer’s suggestion, we have conducted an experiment comparing the performance of residual carrier modulation and the DPT scheme. In the experiment, we used a 3-MHz laser as the signal laser and a 100-kHz laser as the local oscillator. For RCM, the transmitted signal is a dual-subcarrier 45-GBd PS-256-QAM signal. For DPT, the transmitted signal is a 90-GBd PS-256-QAM signal with the same entropy. The setup is the same as Fig. 3 in the manuscript. The results are shown in Fig. R2. For a fair comparison, we employ the pilot tone-to-signal power ratio (PTSPR) at the receiver side as the metric for the power of the pilot tone or residual carrier. PTSPR is defined as

$$PTSPR = 10 \cdot \log_{10}(P_{pilot}/P_{signal}), \tag{R1}$$

where P_{pilot} is the power of the pilot tone component in the electrical spectrum at the receiver, P_{signal} is the power of the signal in the electrical spectrum at the receiver.

We agree that given sufficient DAC resolutions, the performance of the DPT method can approach the residual carrier modulation. It is confirmed in the PTSPR region of [-25 dB, 20 dB] in Fig. R2.

However, since the DPT generates digital pilot tone components at the transmitter through the DAC, it occupies precious DAC quantization bits. Further increasing the PTSPR will increase quantization noise, thereby degrading system performance. It is verified by the turning point around -20 dB PTSPR in the blue curve. In contrast, the residual carrier scheme can further enhance its signal-to-noise ratio and improve performance by increasing the PTSPR, as it is generated in the optical domain. Superior performance with RCM can be observed when the receiver PTSPR is above -13 dB. The NGMI is significantly increased from ~0.84 to ~0.87 compared to the DPT scheme.

Fig. R2. Measured NGMI versus the receiver-side PTSPR for RCM and DPT schemes

Moreover, we also measure the signal performance under different modulation formats with the same experiment setup. As shown in Fig. R3, for low-order modulation formats such as 16-QAM, the DPT can get almost the same phase noise

compensation performance as the RCM scheme. However, for high-order modulation signals, RCM becomes more advantageous over the DPT scheme. For example, a GMI improvement of 0.45 bit/4D-symbol can be obtained for PS-256-QAM with an entropy of 13.92 bit/4D-symbol.

Fig. R3. Measured GMI and GMI difference versus the transmitted signal entropy for RCM and DPT schemes

It is worth noting that we used DACs with 8-bit vertical resolution and an effective number of bits (ENOB) of about 5.5 bits (Keysight 8194A) in this experiment. In practical applications, the resolution of the DACs would likely be lower, making the advantage of RCM even more apparent.

Action: The experimental comparison between RCM and DPT schemes has been added in Supplementary Note 7.2 (The DPT scheme is referred as the RF pilot tone scheme in the revised manuscript).

We have also added Fig. R2 as Fig. 5f in the revised manuscript:

(Main part, page 8, line 5) Fig. 5f compares the phase recovery performance of the proposed RCM and RF pilot tone-based phase recovery. The measured NGMI is presented for the same data rate PS-256-QAM signal, using a 3-MHz signal laser and a 100-kHz LO at the back-to-back scenario with varying receiver-side pilot tone-to-signal power ratios (PTSPR). Receiver-side PTSPR denotes the measured power ratio between the pilot tone and signal in the electrical domain. As RF pilot tone scheme can also track phase continuously, at moderate pilot tone power levels (approximately -20 dB), it exhibits comparable performance to RCM. However, as the RF pilot tone occupies the DAC quantization bits, further increasing its power leads to enhanced quantization noise, thereby degrading NGMI performance. In contrast, RCM scheme can further improve performance by increasing the PTSPR. The peak GMI performance difference between the two schemes is 0.45 bits/4D-symbol (See Supplementary Note 7). Notably, our experiment utilized DACs with 8 bits vertical resolution and an effective number of bits (ENOB) of about 5.5 bits (Keysight 8194A). In practical scenarios with lower DAC resolution, the advantage of RCM would be even more pronounced.

We appreciate the careful review by the reviewers and have modified the manuscript in accordance with their suggestions. Here, we present a point-by-point reply (in blue) to the reviewers' comments (in black), as well as the action taken (in red).

Response to Reviewer #3

General Comment: The authors propose a pilot-tone-assisted phase-noise compensation scheme in digital coherent transmission, especially, for short-haul transmission systems. The author's claim is that the novelty of the authors' proposed technique is that it optically generates pilot tones by detuning modulator bias voltage. However, I have concerns about the novelty of this work. The technique of receiving high multi-level signals with a wide linewidth laser by compensating phase noise with pilot tones has been widely studied since the early days of digital coherent transmission. The modulator bias detuning technique has also been proposed for optical OFDM systems. Therefore, I do not find the novelty and originality in the author's proposed method enough for publication in Nature Communications. However, it may be noteworthy that the authors have refined this technique for modern coherent transmission systems and applied it to high-multi-level signal transmission using probabilistic constellation shaping, achieving notable performance in terms of net rate and laser linewidth. The authors should cite appropriate prior works regarding the techniques utilized in the proposed configuration to further clarify the novelty. Below are my concerns and questions.

Reply: We thank the reviewer for the thorough reading and the valuable suggestions to improve our work. We have carefully studied the reviewers' comments. Point-by-point responses are made as follows.

Comment 1: A method of generating a residual carrier by detuning the modulator bias voltage from the null point as a pilot tone for phase noise compensation has already been proposed in the field of optical OFDM in Ref [1]. Also, the phase noise compensation method using the pilot tone shown in Fig. 2d is almost the same as the method proposed in Ref. [2]. The authors should cite these references for the description of the proposed configuration and clarify the authors' originality with respect to these previous works.

References

[1] S. L. Jansen, I. Morita, and H. Tanaka, "Experimental demonstration of 23.6-Gb/s OFDM with a colorless transmitter," OptoElectronics and Communications Conference (OECC), 2007, PD1-5.

[2] S. L. Jansen, I. Morita, T. C. W. Schenk, N. Takeda, and H. Tanaka, "Coherent Optical 25.8-Gb/s OFDM Transmission Over 4160-km SSMF," J. Lightwave Technol., 26, 1, 6-15, 2008.

Reply: Thanks for your comments. We have added the two pioneering works in the introduction section.

We would like to take this opportunity to highlight the novelty as follows.

(1) This manuscript shows the advantages of the proposed residual carrier modulation (RCM) over **time-domain pilot (TP)**.

a) **Continuous Phase Tracking:** In laser phase noise-dominated scenarios, RCM enables continuous phase tracking, whereas TP schemes can only track pilot symbols periodically at intervals of several dozen symbols. Even with linear interpolation for intermediate values, TP cannot accurately estimate phase at intermediate times.

We conduct a numerical simulation to reveal that RCM provides more accurate phase noise estimation, as shown in Fig. R4. In the simulation, the true phase noise and the phase noise estimated by time-domain pilot or residual carrier, are shown in Fig. R4(a) and Fig. R4(b). The Tx- and Rx-side laser linewidth sum is set as 3 MHz. We use the root mean square error (RMSE) to quantify the deviation between the estimated phase noise $\hat{\phi}(n)$ and the true phase noise $\phi(n)$. RMSE is defined as

$$RMSE = \sqrt{\sum_{n=1}^N |\hat{\phi}(n) - \phi(n)|^2 / N}, \quad (R2)$$

where $\hat{\phi}(n)$ is the estimated phase noise by RCM or TP. $\phi(n)$ is the true phase noise value obtained by dividing the received signal by the transmitted signal.

Fig. R4. The true phase noise and estimated phase noise by (a) TP or (b) RCM.

The calculated RMSE for RCM is 3.54°. In comparison, the calculated RMSE for TP method is 8.85°. Even with linear interpolation, the RMSE for TP remains relatively high at 8.05°.

- b) **Phase SNR Advantage:** In the presence of both phase noise and additive white Gaussian noise (AWGN), the RCM scheme employs a low-pass filter in the phase information extraction process, offering a signal-to-noise ratio (SNR) advantage for phase information estimation. This reduces the impact of mistaking additive noise for phase noise on phase recovery. For RCM, the SNR of the phase information can be described as

$$SNR_{RCM} = P_S/P_N = P_C/(S(f) \cdot B_T). \quad (R3)$$

Here P_S is the power of the phase information signal. P_N is the AWGN noise. P_C is the power of the residual carrier at the receiver. $S(f)$ is the power spectral density (PSD) of the noise. B_T is the low-pass filter bandwidth in RCM.

For traditional time-domain pilot-based phase recovery with a signal bandwidth of B , the SNR of the phase information can be described as

$$SNR_{TP} = P_S/P_N = P_S/(S(f) \cdot B). \quad (R4)$$

Then, it follows that

$$SNR_{RCM}/SNR_{TP} = P_C/P_S \cdot B/B_T = PTSPR \cdot B/B_T. \quad (R5)$$

It can be seen that as long as the ratio of the low-pass filter bandwidth to the signal bandwidth is less than the pilot tone-to-signal power ratio (PTSPR), RCM achieves an SNR advantage over the TP method.

Considering that B_T is determined by the laser linewidth sum, in the early days of digital coherent transmission, the systems are operating at low baud rates. So, the condition where the low-pass filter bandwidth to signal bandwidth ratio is less than PTSPR may not be satisfied, and the performance of RCM and TP may be similar. However, in high-speed systems, the RCM can significantly outperform TP. For example, the PTSPR is about -13 dB in Fig. 5(c), B is around 95 GHz, B_T is 360 MHz, we can get

$$SNR_{RCM}/SNR_{TP} = 10^{-13/10} \cdot \frac{95 \times 10^9}{360 \times 10^6} = 13.2 > 1. \quad (R6)$$

Thus, the condition for SNR advantage holds true.

- (2) The proposed residual carrier modulation does not occupy precious DAC quantization bits, and thus could obtain better performance than conventional **digital pilot-tone (DPT)**, which is also referred as **RF pilot tone** (reference [2]). Here we provide a more comprehensive comparison between DPT and RCM.

In the experiment, we used a 3-MHz laser as the signal laser and a 100-kHz laser as the local oscillator. The results are shown in Fig. R5. For a fair comparison, pilot

tone-to-signal power ratio (PTSPR) at the receiver side is employed as the metric for the power of the pilot tone or residual carrier, which is defined as

$$PTSPR = 10 \cdot \log_{10}(P_{pilot}/P_{signal}) \quad (R7)$$

Here P_{pilot} is the power of the pilot tone component in the electrical spectrum at the receiver, P_{signal} is the power of the signal in the electrical spectrum at the receiver.

When the PTSPR of the DPT is large enough, the performance can be close to that of residual carrier modulation, for PTSPR around -20 dB in Fig. R5.

However, since the DPT generates digital pilot tone components at the transmitter through the DAC, it occupies precious DAC quantization bits. Further increasing the PTSPR will increase quantization noise, thereby degrading system performance. In contrast, the residual carrier scheme can further improve performance by increasing the PTSPR. A much better performance with RCM can be observed when the receiver PTSPR is above -13 dB, compared to DPT scheme.

Moreover, we also measured the signal performance under different modulation formats with the same experiment setup. As shown in Fig. R6, for low-order modulation formats such as 16-QAM, the DPT can get almost the same phase noise compensation performance as the RCM scheme. However, for high-order modulation signals, RCM can get a better performance. For example, a GMI improvement of 0.45 bit/4D-symbol can be obtained for PS-256-QAM with an entropy of 13.92 bit/4D-symbol.

It is worth noting that our experiment used DACs with 8-bit vertical resolution and an effective number of bits (ENOB) of about 5.5 bits (Keysight 8194A). In practical applications, the resolution of the DACs would likely be lower, making the advantage of RCM even more apparent.

Fig. R5. Measured NGMI versus the receiver-side PTSPR for RCM and DPT schemes

Fig. R6. Measured GMI and GMI difference versus the transmitted signal entropy for RCM and DPT schemes

- (3) Compared to the OFDM modulation (reference [1] and reference [2]), we proposed the integration of subcarrier modulation and optical residual carrier. As the influence of laser phase noise is characterized by the product of $\Delta f \cdot T_S$, in OFDM systems, low-speed parallel signal with long symbol periods (proportional to the FFT points) is more susceptible to phase noise. On the other hand, in the subcarrier systems, high-speed serial signals with much shorter symbol periods can tolerate larger laser linewidths.
- (4) This manuscript demonstrates a record 1.0-Tb/s PS-256-QAM signal transmission with a 3-MHz linewidth DFB laser. The signal-transparent processing also allows high-order analogue radio-over-fibre applications in fibre-wireless converged networks. It paves the way for the deployment of low-cost DFB lasers with MHz linewidth in commercial short-reach communication systems with minimal hardware complexity.

Action: We have added the two pioneering works [1, 2] in the introduction section: **(Main part, page 2, line 35)** Alternatively, a radio-frequency (RF) pilot tone-based phase recovery technique [24, 25] is proposed, but it consumes precious quantization bits of DACs, enhancing the quantization noise and degrading signal SNR. Although an optical generation of pilot tone has been demonstrated with OFDM modulation [26], OFDM is much more susceptible to phase noise due to its longer symbol duration. Besides, the potential broad application of optical pilot tone in modern optical communication remains unexplored. Furthermore, overcoming phase noise in high-order modulation standard coherent systems with low-cost lasers has not yet been demonstrated.

(Main part, page 4, line 59) An alternative phase recovery scheme involves utilizing a radio-frequency pilot tone [24, 25], as depicted in Fig. 2f.

The experimental comparison between RCM and DPT schemes has also been added

in Supplementary Note 7.2. (The DPT scheme is referred as RF pilot tone scheme to distinguish from the optical-domain RCM scheme in the revised manuscript).

We have also added Fig. R5 as Fig. 5f in the revised manuscript:

(Main part, page 8, line 5) Fig. 5f compares the phase recovery performance of the proposed RCM and RF pilot tone-based phase recovery. The measured NGMI is presented for the same data rate PS-256-QAM signal, using a 3-MHz signal laser and a 100-kHz LO at the back-to-back scenario with varying receiver-side pilot tone-to-signal power ratios (PTSPR). Receiver-side PTSPR denotes the measured power ratio between the pilot tone and signal in the electrical domain. As RF pilot tone scheme can also track phase continuously, at moderate pilot tone power levels (approximately -20 dB), it exhibits comparable performance to RCM. However, as the RF pilot tone occupies the DAC quantization bits, further increasing its power leads to enhanced quantization noise, thereby degrading NGMI performance. In contrast, RCM scheme can further improve performance by increasing the PTSPR. The peak GMI performance difference between the two schemes is 0.45 bits/4D-symbol (See Supplementary Note 7). Notably, our experiment utilized DACs with 8 bits vertical resolution and an effective number of bits (ENOB) of about 5.5 bits (Keysight 8194A). In practical scenarios with lower DAC resolution, the advantage of RCM would be even more pronounced.

Comment 2: I think the intrinsic comparison between the proposed method and the conventional methods (RF tone method and TP method) is insufficient. Detuning the bias voltage causes distortion in the signal constellation. According to the supplements, I guess that the authors apply a Volterra nonlinear equalizer to the receiver side to deal with this, but it is not stated clearly. The bias detuning penalty and how it is addressed should be explained for comparison with conventional RF pilot tone schemes. Are the benefits of the proposed scheme sufficient compared to the high cost and complexity of DSP due to the Volterra nonlinear equalizer?

Reply: Thanks for your insightful comment.

- (1) We compared our proposed RCM with the RF tone method in the reply to Comment #1, demonstrating that RCM achieves better performance without occupying DAC quantization bits.
- (2) Indeed, deviation from the extinction point may lead to a decrease in modulation linearity. We add the NGMI result using linear equalization only in Fig. R7. The equalizer taps of different systems are optimized respectively to ensure a fair comparison.

The NGMI difference between linear and Volterra nonlinear equalization of RCM is shown in Fig. R8. It can be shown that the NGMI difference between linear equalization and Volterra nonlinear equalization is quite small. This observation is attributed to the relatively low PTSPR of the RCM. Therefore, the results reveal the performance and power-efficient advantages of the RCM method.

For computational complexity, we use a sparse Volterra nonlinear equalizer with diagonal terms only. The typical taps lengths of the 1st-, 2nd-, 3rd-kernels are 80, 20, 5, respectively. Therefore, the number of multiplications is increased by ~ 69% compared with linear FFE.

Fig. R7. Measured NGMI versus receiver-side PTSPR for RCM and DPT schemes with (w/) and without (w/o) VNLE

Fig. R8. Measured NGMI difference versus receiver-side PTSPR for RCM with (w/) and without (w/o) VNLE

Action: We have added the linear and nonlinear equalization performance of RCM in Supplementary Note 8.

Comment 3: In a typical digital coherent transmitter, an auto-bias control is used to tune the bias of the IQ modulator to the null point. How did the authors achieve a stable bias voltage detuned from the null point?

Reply: Thanks for your comments. In our experiment, we manually adjust the bias voltage to generate a residual carrier with a target PTSPR.

From the perspective of practical use, the most common auto-bias control methods for high-order modulation formats are the dither signal detection techniques [R2, R3, R4]. These methods introduce small, low-frequency sine or square waves as the dither signal into the bias pin of the modulator, and adjust the bias voltage automatically according to the output optical signal.

By using dither-correlation detection and a specifically designed dither signal loading scheme [R4], precise and stable arbitrary bias point control can be achieved. Therefore, this technique is feasible to maintain a stable bias voltage in residual carrier modulation systems.

[R2] H. Kawakami, T. Kobayashi, E. Yoshida, and Y. Miyamoto, "Auto bias control technique for optical 16-QAM transmitter with asymmetric bias dithering," *Opt. Express*, vol. 19, no. 26, pp. B308-B312, 2011.

[R3] X. Zhu et al., "Coherent Detection-Based Automatic Bias Control of Mach–Zehnder Modulators for Various Modulation Formats," *J. Lightw. Technol.*, vol. 32, no. 14, pp. 2502-2509, 2014.

[R4] X. Li et al., "Arbitrary Bias Point Control Technique for Optical IQ Modulator Based on Dither-Correlation Detection," *J. Lightw. Technol.*, vol. 36, no. 18, pp. 3824-3836, 2018.

Comment 4: I guess that the residual carrier component power after polarization splitting in a polarization diversity receiver is fluctuating due to polarization rotation and polarization-mode dispersion in the transmission line. That is, CSPR will be fluctuating. This is caused by interference between the residual carrier components between orthogonal polarizations. How did the authors manage to address this issue?

Reply: We would like to thank the reviewer for the valuable comment.

We agree that the CSPR on each polarization fluctuates with polarization rotation in the fibre link, due to the interference between the residual carrier components between orthogonal polarizations. In our manuscript, we address this issue by recovering the phase on the residual carrier at the polarization with a higher CSPR. Here we name it the **selection combining method (SC)**. Similar to wireless communications, despite the SC we used, other methods such as **equal gain combining (EGC)** and **maximum ratio combining (MRC)** can also be utilized for combining the phase information from residual carriers on two orthogonal polarizations. The main principle is described as follows:

- (1) **Selection combining method (SC):** select the residual component that has higher CSPR in the two polarizations.
- (2) **Equal gain combining method (EGC):** add the two residual components with equal weights.

(3) **Maximum ratio combining method (MRC):** add the two residual components with different weights, proportional to their CSPR.

Fig. R9. Measured NGMI under different RSOP with (a) selection combining, (b) equal gain combining, and (c) maximum ratio combining strategies.

For the EGC and MRC, when a phase difference of π occurs between the residual carrier components between orthogonal polarizations due to the polarization rotation, the fixed phase difference is detected and compensated first before combining.

We conduct simulations to compare the performance of these three strategies.

Polarization rotation is simulated by traversing the rotation of state of polarization (RSOP) as follows [R5]

$$\begin{bmatrix} E_{x,out} \\ E_{y,out} \end{bmatrix} = \begin{bmatrix} \cos\theta & e^{-j\phi}\sin\theta \\ -e^{j\phi}\sin\theta & \cos\theta \end{bmatrix} \begin{bmatrix} E_{x,in} \\ E_{y,in} \end{bmatrix}. \quad (R8)$$

Here $E_{x,in}$ and $E_{y,in}$ is the Jones vector components of the optical field at the transmitter. $E_{x,out}$ and $E_{y,out}$ is the Jones vector components at the receiver.

The simulation system uses a 3-MHz linewidth laser as the signal laser and a 100-kHz linewidth laser as the LO. The transmitted signal is dual-subcarrier 45-GBd PS-256-QAM, which is consistent with the manuscript.

As shown in Fig. R9, the measured NGMI is observed to be stable under all different RSOPs, and the three strategies exhibit similar performance.

The mean value and standard deviation of measured NGMI using different combining strategies are shown in Fig. R10. It can be seen that MRC can get slightly better performance. However, the difference is quite small to ensure the practical use of SC, EGC, and MRC.

Fig. R10. Average NGMI and standard deviation of NGMI with different combining strategies

Regarding the polarization-mode-dispersion (PMD) effect, the standard single-mode fibre has a PMD coefficient of 0.2ps/sqrt(km). Even without considering polarization rotation (the worst case), the PMD-induced differential delay at 80 km is 1.79 ps. Additionally, since phase noise is narrowband (typically several hundred MHz), this delay can be considered as introducing phase modulation $\phi = \exp(j2\pi f\tau)$ on a particular polarization. Therefore, the impact of this delay can be ignored.

[R5] S. J. Savory, "Digital Coherent Optical Receivers: Algorithms and Subsystems," in IEEE Journal of Selected Topics in Quantum Electronics, vol. 16, no. 5, pp. 1164-1179, 2010.

Action: The simulation comparison among selection combining (SC), equal gain combining (EGC), and (3) Maximum ratio combining (MRC) over various rotations of state of polarization (RSOP) have been added in Supplementary Note 9.

REVIEWERS' COMMENTS

Reviewer #1 (Remarks to the Author):

This paper describes the use of residual carrier modulation (RCM) to reduce laser phase noise and employ low-cost lasers. Their technical implantation has shown remarkable performance. By adjusting the bias point of the IQ modulator, the residual carrier can be effectively used to track the phase changes. The authors simulated and compared the performance for the PS-256-QAM signals with TP-based phase recovery and showed the superior performance of the RCM system. They also demonstrated the experimental results using a high-performance oscilloscope. It is a good way to utilize the residual carrier to minimize the cost and the system requirement. They achieved 1.0 Tb/s PS-256-QAM transmission with a 3 MHz linewidth laser, which is a very useful result with a simple, low-cost system. The long-term stability of the bias point can also be secured by the dither detection-based auto-bias control technique. The manuscript seems to be well prepared and it could be published.

Reviewer #3 (Remarks to the Author):

The authors have addressed the reviewers' concerns sincerely. A detailed discussion was added to the supplement regarding my technical questions (non-linearity of modulation and the effects of polarization variations in tone). I recommend that a discussion of bias adjustment, which many researchers in this area may question, also be added either in the supplement or in the main body. Meanwhile, I still have a concern about the statement of the novelty of the authors' proposal. The authors state that OFDM is susceptible to phase noise as the difference between the authors' proposal and a previous study already proposed in the context of OFDM. I agree with this, but this only indicates that OFDM applications have low phase noise tolerance. This cannot be an argument that the authors' scheme is new to the schemes proposed in the previous study for OFDM applications. The authors' proposal and novelty are the combination of optical carrier (optical tone) generation by bias shifting (this basic concept has already been proposed in Ref. [26]) and subcarrier multiplexing of high-speed signals in modern optical communication systems, the quantitative evaluation of the superiority of phase tracking with TP, and the achievement of remarkable performance in terms of net rate and laser linewidth. The authors should make this clear in the introduction. In page 3, left column, line 12, although the authors state "In this work, we propose a residual carrier modulation technique as a cost-effective solution for phase noise mitigation.", this statement should be corrected.

Reviewer#3 feedback on reviewer#2 and response letter is as below.

With respect to Reviewer 2's Comment 1, we believe the authors' explanation on their response letter is reasonable. As mentioned by authors, the DC carrier is actually not likely to be lost in the receiver analog circuit because of the frequency offset. However, in view of the possibility that readers may have questions about this, I think it should be mentioned in the paper in some way.

In Comment 2, as the authors say, I see this comment contains a misunderstanding. In this paper, the authors mainly compare a method that use a tone signal and a method that use time-embedded pilot symbols. This is not a comparison between a digital tone technique and an analog tone technique. The additional data in Fig. 5f, which compares the digital and analog approaches that Reviewer 2 was concerned about, is good data to show the effectiveness of the authors' proposed method.

We appreciate the careful review by the reviewers and have modified the manuscript in accordance with their suggestions. Here, we present a point-by-point reply (in blue) to the reviewers' comments (in black), as well as the action taken (in red).

Response to Reviewer #1

Comment: This paper describes the use of residual carrier modulation (RCM) to reduce laser phase noise and employ low-cost lasers. Their technical implantation has shown remarkable performance. By adjusting the bias point of the IQ modulator, the residual carrier can be effectively used to track the phase changes. The authors simulated and compared the performance for the PS-256-QAM signals with TP-based phase recovery and showed the superior performance of the RCM system. They also demonstrated the experimental results using a high-performance oscilloscope. It is a good way to utilize the residual carrier to minimize the cost and the system requirement. They achieved 1.0 Tb/s PS-256-QAM transmission with a 3 MHz linewidth laser, which is a very useful result with a simple, low-cost system. The long-term stability of the bias point can also be secured by the dither detection-based auto-bias control technique. The manuscript seems to be well prepared and it could be published.

Reply: We would like to express our gratitude to the reviewer for the positive feedback.

Action: We have added the description of the dither detection-based auto-bias control technique as a potential solution in Supplementary Note 10.

We appreciate the careful review by the reviewers and have modified the manuscript in accordance with their suggestions. Here, we present a point-by-point reply (in blue) to the reviewers' comments (in black), as well as the action taken (in red).

Response to Reviewer #3

Comment 1: The authors have addressed the reviewers' concerns sincerely. A detailed discussion was added to the supplement regarding my technical questions (non-linearity of modulation and the effects of polarization variations in tone). I recommend that a discussion of bias adjustment, which many researchers in this area may question, also be added either in the supplement or in the main body.

Reply: We would like to thank the reviewer for the constructive advice.

Action: We have added the discussion of bias adjustment and related references to the Supplementary Note 10.

Comment 2: Meanwhile, I still have a concern about the statement of the novelty of the authors' proposal. The authors state that OFDM is susceptible to phase noise as the difference between the authors' proposal and a previous study already proposed in the context of OFDM. I agree with this, but this only indicates that OFDM applications have low phase noise tolerance. This cannot be an argument that the authors' scheme is new to the schemes proposed in the previous study for OFDM applications. The authors' proposal and novelty are the combination of optical carrier (optical tone) generation by bias shifting (this basic concept has already been proposed in Ref. [26]) and subcarrier multiplexing of high-speed signals in modern optical communication systems, the quantitative evaluation of the superiority of phase tracking with TP, and the achievement of remarkable performance in terms of net rate and laser linewidth. The authors should make this clear in the introduction. In page 3, left column, line 12, although the authors state "In this work, we propose a residual carrier modulation technique as a cost-effective solution for phase noise mitigation.", this statement should be corrected.

Reply: We would like to thank the reviewer for the correction and summarization of the contribution of our work.

We have revised the statements about the novelty in the Introduction. Our solution to overcome phase noise is straightforward, which does not add extra hardware complexity, but obtains significant performance improvement. Therefore, we are very confident in this solution.

Action: The statement of the novelty in the Introduction is revised in the manuscript: **(Main part, page 2, line 36)** In this work, we present a residual carrier modulation technique integrated with subcarrier multiplexing to achieve high-speed and high spectral efficiency in modern coherent optical communication systems using low-cost, large linewidth lasers. Our quantitative evaluation demonstrates the advantage of our phase tracking method over conventional time-domain and frequency-domain pilot techniques. We also showcase its potential in diverse scenarios such as data-center interconnects and fiber-wireless access networks.

We appreciate the careful review by the reviewers and have modified the manuscript in accordance with their suggestions. Here, we present a point-by-point reply (in blue) to the reviewers' comments (in black), as well as the action taken (in red).

Response to Reviewer#3 Feedback on Reviewer#2

Comment 1: With respect to Reviewer 2's Comment 1, we believe the authors' explanation on their response letter is reasonable. As mentioned by authors, the DC carrier is actually not likely to be lost in the receiver analog circuit because of the frequency offset. However, in view of the possibility that readers may have questions about this, I think it should be mentioned in the paper in some way.

Action: The explanation for the DC block problem is added in the revised manuscript: **(Main part, page 2, line 90)** It should be noted that the frequency offset prevents the residual carrier from being blocked in the receiver analog circuit.

Comment 2: In Comment 2, as the authors say, I see this comment contains a misunderstanding. In this paper, the authors mainly compare a method that use a tone signal and a method that use time-embedded pilot symbols. This is not a comparison between a digital tone technique and an analog tone technique. The additional data in Fig. 5f, which compares the digital and analog approaches that Reviewer 2 was concerned about, is good data to show the effectiveness of the authors' proposed method.

Reply: We would like to express our gratitude to the reviewer for the positive feedback.